# 3D Equivariant Diffusion for Target-Aware Molecule Generation and Affinity Prediction

**Jiaqi Guan**[1]*, **Wesley Wei Qian**[1]*, **Xingang Peng**[2], **Yufeng Su**[1], **Jian Peng**[1], **Jianzhu Ma**[3]

[1] Department of Computer Science, University of Illinois Urbana-Champaign
[2] School of Intelligence Science and Technology, Peking University
[3] Institute for AI industry Research, Tsinghua University
{jiaqi, weiqian3, jianpeng}@illinois.edu,
majianzhu@air.tsinghua.edu.cn

## Abstract

Rich data and powerful machine learning models allow us to design drugs for a specific protein target *in silico*. Recently, the inclusion of 3D structures during targeted drug design shows superior performance to other target-free models as the atomic interaction in the 3D space is explicitly modeled. However, current 3D target-aware models either rely on the voxelized atom densities or the autoregressive sampling process, which are not equivariant to rotation or easily violate geometric constraints resulting in unrealistic structures. In this work, we develop a 3D equivariant diffusion model to solve the above challenges. To achieve target-aware molecule design, our method learns a joint generative process of both continuous atom coordinates and categorical atom types with a SE(3)-equivariant network. Moreover, we show that our model can serve as an unsupervised feature extractor to estimate the binding affinity under proper parameterization, which provides an effective way for drug screening. To evaluate our model, we propose a comprehensive framework to evaluate the quality of sampled molecules from different dimensions. Empirical studies show our model could generate molecules with more realistic 3D structures and better affinities towards the protein targets, and improve binding affinity ranking and prediction without retraining.

## 1 Introduction

Rational drug design against a known protein binding pocket is an efficient and economical approach for finding lead molecules (Anderson, 2003; Batool et al., 2019) and has attracted growing attention from the research community. However, it remains challenging and computationally intensive due to the large synthetically feasible space (Ragoza et al., 2022), and high degrees of freedom for binding poses (Hawkins, 2017). Previous prevailed molecular generative models are based on either molecular string representation (Bjerrum and Threlfall, 2017; Kusner et al., 2017; Segler et al., 2018) or graph representation (Li et al., 2018; Liu et al., 2018; Jin et al., 2018; Shi et al., 2020), but both representations do not take the 3D spatial interaction into account and therefore not well suited for *target-aware* molecule generation. With recent development in structural biology and protein structure prediction (Jumper et al., 2021), more structural data become available (Francoeur et al., 2020) and unlock new opportunities for machine learning algorithms to directly design drugs inside 3D binding complex (Gebauer et al., 2019; Simm et al., 2020a;b).

Recently, new generation of generative models are proposed specifically for the target-aware molecule generation task (Luo et al., 2021; Ragoza et al., 2022; Tan et al., 2022; Liu et al., 2022; Peng et al., 2022). However, existing approaches suffer from several drawbacks. For instance, Tan et al. (2022) does not explicitly model the interactions between atoms of molecules and proteins in the 3D space, but only considers the target as intermediate conditional embeddings. For those that do consider the atom interactions in the 3D space, Ragoza et al. (2022) represents the 3D space as voxelized grids and model the proteins and molecules using 3D Convolutional Neural Networks (CNN). However, this model is not rotational equivariant and cannot fully capture the 3D inductive

---

*Equal Contribution

biases. In addition, the voxelization operation will lead to poor scalability since the number of voxels increases at a cubic rate to the pocket size. Advanced approaches achieve SE(3)-equivariance through different modeling techniques (Luo et al., 2021; Liu et al., 2022; Peng et al., 2022). However, these methods adopt autoregressive sampling, where atoms are generated one by one based on the learned probability density of atom types and atom coordinates. These approaches suffer from several limitations: First, the mismatch between training and sampling incurs exposure bias. Secondly, the model assigns an unnatural generation order during sampling and cannot consider the probability of the entire 3D structure. For instance, it would be easy for the model to correctly place the $n$-th atom to form a benzene ring if the $n-1$-th carbon atoms have already been placed in the same plane. However, it would be difficult for the model to place the first several atoms accurately since there is limited context information available, which yields unrealistic fragments as a consequence. Moreover, the sampling scheme does not scale well when generating large binding molecules is necessary. Finally, current autoregressive models could not estimate the quality of generated molecules. One has to rely on other tools based on physical-chemical energy functions such as AutoDock (Trott and Olson, 2010) to select the drug candidates.

To address these problems, we propose TargetDiff, a 3D full-atom diffusion model that generates target-aware molecules in a non-autoregressive fashion. Thanks to recent progress in probabilistic diffusion models (Ho et al., 2020; Hoogeboom et al., 2021) and equivariant neural networks (Fuchs et al., 2020; Satorras et al., 2021b), our proposed model can generate molecules in continuous 3D space based on the context provided by protein atoms, and have the invariant likelihood w.r.t global translation and rotation of the binding complex. Specifically, we represent the protein binding pockets and small molecules as atom point sets in the 3D space where each atom is associated with a 3D Cartesian coordinate. We define a diffusion process for both *continuous* atom coordinates and *discrete* atom types where noise is gradually added, and learn the joint generative process with a SE(3)-equivariant graph neural network which alternately updates the atom hidden embedding and atom coordinates of molecules. Under certain parameterization, we can extract representative features from the model by forward passing the input molecules once without retraining. We find these features provide strong signals to estimate the binding affinity between the sampled molecule and target protein, which can then be used for ranking drug candidates and improving other supervised learning frameworks for binding affinity prediction. An empirical study on the CrossDocked2020 dataset (Francoeur et al., 2020) shows that TargetDiff generates molecules with more realistic 3D structures and better binding energies towards the protein binding sites compared to the baselines.

Our main contributions can be summarized as follows:

- An end-to-end framework for generating molecules conditioned on a protein target, which explicitly considers the physical interaction between proteins and molecules in 3D space.

- So far as we know, this is the first probabilistic diffusion formulation for *target-aware* drug design, where training and sampling procedures are aligned in a non-autoregressive as well as SE(3)-equivariant fashion thanks to a shifting center operation and equivariant GNN.

- Several new evaluation metrics and additional insights that allow us to evaluate the model generated molecules in many different dimensions. The empirical results demonstrate the superiority of our model over two other representative baselines.

- Propose an effective way to evaluate the quality of generated molecules based on our framework, where the model can be served as either a scoring function to help ranking or an unsupervised feature extractor to improve binding affinity prediction.

## 2 RELATED WORK

**Molecule Generation with Different Representations**   Based on different levels of representations, existing molecular generative models can be roughly divided into three categories - string-based, graph-based, and 3D-structure-based. The most common molecular string representation is SMILES (Weininger, 1988), where many existing language models such as RNN can be re-purposed for the molecule generation task (Bjerrum and Threlfall, 2017; Gómez-Bombarelli et al., 2018; Kusner et al., 2017; Segler et al., 2018). However, SMILES representation is not an optimal choice since it fails to capture molecular similarities and suffers from the validity issue during the generation phase (Jin et al., 2018). Thus, many graph-based methods are proposed to operate directly on

graphs (Liu et al., 2018; Shi et al., 2020; Jin et al., 2018; 2020; You et al., 2018; Zhou et al., 2019). On the other hand, these methods are very limited in modeling the spatial information of molecules that is crucial for determining molecular properties and functions. Therefore, recent work (Gebauer et al., 2019; Skalic et al., 2019a; Ragoza et al., 2020; Simm et al., 2020a;b) focus on generating molecules in 3D space. More recently, flow-based and diffusion-based generative models (Satorras et al., 2021a; Hoogeboom et al., 2022) are developed to leverage E(n)-Equivariant GNN (Satorras et al., 2021b) and achieve SE(3)-equivariance in molecule generation.

**Target-Aware Molecule Generation**  As more structural data become available, various generative models are proposed to solve the *target-aware* molecule generation task. For example, Skalic et al. (2019b); Xu et al. (2021) generate SMILES based on protein contexts. Tan et al. (2022) propose a flow-based model to generate molecular graphs conditional on a protein target as a sequence embedding. Ragoza et al. (2022) try to generate 3D molecules by voxelizing molecules in atomic density grids in a conditional VAE framework. Li et al. (2021) leverage Monte-Carlo Tree Search and a policy network to optimize molecules in 3D space. Luo et al. (2021); Liu et al. (2022); Peng et al. (2022) develop autoregressive models to generate molecules atom by atom in 3D space with GNNs. Despite the progress made in this direction, the models still suffer from several issues, including separately encoding the small molecules and protein pockets (Skalic et al., 2019b; Xu et al., 2021; Tan et al., 2022; Ragoza et al., 2022), relying on voxelization and non-equivariance networks (Skalic et al., 2019b; Xu et al., 2021; Ragoza et al., 2022), and autoregressive sampling (Luo et al., 2021; Liu et al., 2022; Peng et al., 2022). Different from all these models, our equivariant model explicitly considers the interaction between proteins and molecules in 3D and can perform non-autoregressive sampling, which better aligns the training and sampling procedures.

**Diffusion Models**  Diffusion models (Sohl-Dickstein et al., 2015) are a new family of latent variable generative models. Ho et al. (2020) propose denoising diffusion probabilistic models (DDPM) which establishes a connection between diffusion models and denoising score-based models (Song and Ermon, 2019). The diffusion models have shown remarkable success in generating image data (Ho et al., 2020; Nichol and Dhariwal, 2021) and discrete data such as text (Hoogeboom et al., 2021; Austin et al., 2021). Recently, it has also been applied in the domain of molecules. For example, GeoDiff (Xu et al., 2022) generates molecular conformations given 2D molecular graphs. EDM (Hoogeboom et al., 2022) generates 3D molecules. However, the unawareness to potential targets make it hard to be utilized by biologists in real scenarios.

## 3 METHODS

### 3.1 PROBLEM DEFINITION

A protein binding site is represented as a set of atoms $\mathcal{P} = \{(\boldsymbol{x}_P^{(i)}, \boldsymbol{v}_P^{(i)})\}_{i=1}^{N_P}$, where $N_P$ is the number of protein atoms, $\boldsymbol{x}_P \in \mathbb{R}^3$ represents the 3D coordinates of the atom, and $\boldsymbol{v}_P \in \mathbb{R}^{N_f}$ represents protein atom features such as element types and amino acid types. Our goal is to generate binding molecules $\mathcal{M} = \{(\boldsymbol{x}_L^{(i)}, \boldsymbol{v}_L^{(i)})\}_{i=1}^{N_M}$ conditioned on the protein target. For brevity, we denote molecules as $M = [\mathbf{x}, \mathbf{v}]$, where $[\cdot, \cdot]$ is the concatenation operator and $\mathbf{x} \in \mathbb{R}^{M \times 3}$ and $\mathbf{v} \in \mathbb{R}^{M \times K}$ denote atom Cartesian coordinates and one-hot atom types respectively.

### 3.2 OVERVIEW OF TARGETDIFF

As discussed in Sec. 1 and Sec. 2, we hope to develop a non-autoregressive model to bypass the drawbacks raised in autoregressive sampling models. In addition, we also require the model to represent the protein-ligand complex in continuous 3D space to avoid the voxelization operation. Last but not least, the model will also need to be SE(3)-equivariant to global translation and rotation.

We therefore develop TargetDiff, an equivariant non-autoregressive method for target-aware molecule generation based on the DDPM framework Ho et al. (2020). TargetDiff is a latent variable model of the form $p_\theta(M_0|\mathcal{P}) = \int p_\theta(M_{0:T}|\mathcal{P})dM_{1:T}$, where $M_1, M_2, \cdots, M_T$ is a sequence of latent variables with the same dimensionality as the data $M_0 \sim p(M_0|\mathcal{P})$. As shown in Fig. 1, the approach includes a forward *diffusion* process and a reverse *generative* process, both defined as Markov chains. The diffusion process gradually injects noise to data, and the generative process

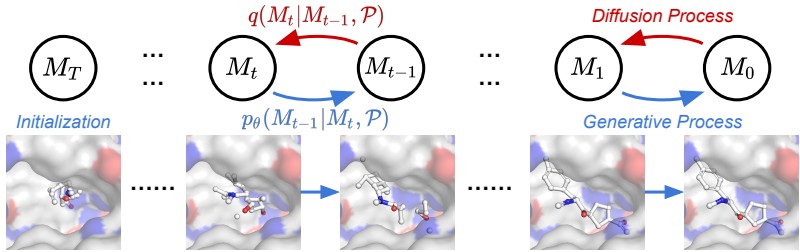

**Figure 1:** Overview of TargetDiff. The diffusion process gradually injects noise to the data, and the generative process learns to recover the data distribution from the noise distribution with a network parameterized by $\theta$.

learns to recover data distribution from the noise distribution with a network parameterized by $\theta$:

$$q(M_{1:T}|M_0, \mathcal{P}) = \Pi_{t=1}^T q(M_t|M_{t-1}, \mathcal{P}) \qquad p_\theta(M_{0:T-1}|M_T, \mathcal{P}) = \Pi_{t=1}^T p_\theta(M_{t-1}|M_t, \mathcal{P}) \quad (1)$$

Since our goal is to generate 3D molecules based on a given protein binding site, the model needs to generate both *continuous* atom coordinates and *discrete* atom types, while keeping SE(3)-equivariant during the entire generative process. In the following section, we will elaborate on how we construct the diffusion process, parameterize the generative process, and eventually train the model.

### 3.3 MOLECULAR DIFFUSION PROCESS

Following recent progress in learning continuous distributions Ho et al. (2020) and discrete distributions Hoogeboom et al. (2021) with diffusion models, we use a Gaussian distribution $\mathcal{N}$ to model continuous atom coordinates $\mathbf{x}$ and a categorical distribution $\mathcal{C}$ to model discrete atom types $\mathbf{v}$. The atom types are constructed as a one-hot vector containing information such as element types and membership in an aromatic ring. We formulate the molecular distribution as a product of atom coordinate distribution and atom type distribution. At each time step $t$, a small Gaussian noise and a uniform noise across all categories are added to atom coordinates and atom types separately, according to a Markov chain with fixed variance schedules $\beta_1, \ldots, \beta_T$:

$$q(M_t|M_{t-1}, \mathcal{P}) = \mathcal{N}(\mathbf{x}_t; \sqrt{1-\beta_t}\mathbf{x}_{t-1}, \beta_t \mathbf{I}) \cdot \mathcal{C}(\mathbf{v}_t|(1-\beta_t)\mathbf{v}_{t-1} + \beta_t/K). \quad (2)$$

We note that the schedules can be different in practice, but we still denote them with the same symbol for conciseness. Here, we decompose the joint molecule distribution as the product of two independent distributions of atom coordinates and atom types during diffusion, because the independent distributions have concise mathematical formulations and we can efficiently draw noisy samples from them. In the next section, we will see the dependencies between atom coordinates and atom types are considered by the model in the generative process.

Denoting $\alpha_t = 1 - \beta_t$ and $\bar{\alpha}_t = \Pi_{s=1}^t \alpha_s$, a desirable property of the diffusion process is to calculate the noisy data distribution $q(M_t|M_0)$ of any time step in closed-form:

$$q(\mathbf{x}_t|\mathbf{x}_0) = \mathcal{N}(\mathbf{x}_t; \sqrt{\bar{\alpha}_t}\mathbf{x}_0, (1-\bar{\alpha}_t)\mathbf{I}) \qquad q(\mathbf{v}_t|\mathbf{v}_0) = \mathcal{C}(\mathbf{v}_t|\bar{\alpha}_t\mathbf{v}_0 + (1-\bar{\alpha}_t)/K). \quad (3)$$

Using Bayes theorem, the normal posterior of atom coordinates and categorical posterior of atom types can both be computed in closed-form:

$$q(\mathbf{x}_{t-1}|\mathbf{x}_t, \mathbf{x}_0) = \mathcal{N}(\mathbf{x}_{t-1}; \tilde{\boldsymbol{\mu}}_t(\mathbf{x}_t, \mathbf{x}_0), \tilde{\beta}_t \mathbf{I}) \qquad q(\mathbf{v}_{t-1}|\mathbf{v}_t, \mathbf{v}_0) = \mathcal{C}(\mathbf{v}_{t-1}|\tilde{\boldsymbol{c}}_t(\mathbf{v}_t, \mathbf{v}_0)). \quad (4)$$

where $\tilde{\boldsymbol{\mu}}_t(\mathbf{x}_t, \mathbf{x}_0) = \frac{\sqrt{\bar{\alpha}_{t-1}}\beta_t}{1-\bar{\alpha}_t}\mathbf{x}_0 + \frac{\sqrt{\alpha_t}(1-\bar{\alpha}_{t-1})}{1-\bar{\alpha}_t}\mathbf{x}_t$, $\tilde{\beta}_t = \frac{1-\bar{\alpha}_{t-1}}{1-\bar{\alpha}_t}\beta_t$,

and $\tilde{\boldsymbol{c}}_t(\mathbf{v}_t, \mathbf{v}_0) = \boldsymbol{c}^\star / \sum_{k=1}^K c_k^\star$ and $\boldsymbol{c}^\star(\mathbf{v}_t, \mathbf{v}_0) = [\alpha_t \mathbf{v}_t + (1-\alpha_t)/K] \odot [\bar{\alpha}_{t-1}\mathbf{v}_0 + (1-\bar{\alpha}_{t-1})/K]$.

### 3.4 PARAMETERIZATION OF EQUIVARIANT MOLECULAR GENERATIVE PROCESS

The generative process, on reverse, will recover the ground truth molecule $M_0$ from the initial noise $M_T$, and we approximate the reverse distribution with a neural network parameterized by $\theta$:

$$p_\theta(M_{t-1}|M_t, \mathcal{P}) = \mathcal{N}(\mathbf{x}_{t-1}; \boldsymbol{\mu}_\theta([\mathbf{x}_t, \mathbf{v}_t], t, \mathcal{P}), \sigma_t^2 I) \cdot \mathcal{C}(\mathbf{v}_{t-1}|\boldsymbol{c}_\theta([\mathbf{x}_t, \mathbf{v}_t], t, \mathcal{P})). \quad (5)$$

One desired property of the generative process is that the likelihood $p_\theta(M_0|\mathcal{P})$ should be invariant to translation and rotation of the protein-ligand complex, which is a critical inductive bias for generating 3D objects such as molecules (Köhler et al., 2020; Satorras et al., 2021a; Xu et al., 2022; Hoogeboom et al., 2022). One important piece of evidence for such achievement is that an invariant distribution composed with an equivariant transition function will result in an invariant distribution. Leveraging this evidence, we have the following proposition in the setting of target-aware molecule

**Proposition 1.** *Denoting the SE(3)-transformation as $T_g$, we could achieve invariant likelihood w.r.t $T_g$ on the protein-ligand complex: $p_\theta(T_g(M_0|\mathcal{P})) = p_\theta(M_0|\mathcal{P})$ if we shift the Center of Mass (CoM) of protein atoms to zero and parameterize the Markov transition $p(\mathbf{x}_{t-1}|\mathbf{x}_t, \mathbf{x}_\mathcal{P})$ with an SE(3)-equivariant network.*

A slight abuse of notation in the following is that we use $\mathbf{x}_t(t = 1, \ldots, T)$ to denote ligand atom coordinates and $\mathbf{x}_\mathcal{P}$ to denote protein atom coordinates. We analyze the operation of shifting CoM in Appendix B and prove the invariant likelihood in Appendix C.

There are different ways to parameterize $\boldsymbol{\mu}_\theta([\mathbf{x}_t, \mathbf{v}_t], t, \mathcal{P})$ and $\boldsymbol{c}_\theta([\mathbf{x}_t, \mathbf{v}_t], \mathcal{P})$. Here, we choose to let the neural network predict $[\mathbf{x}_0, \mathbf{v}_0]$ and feed it through equation 4 to obtain $\boldsymbol{\mu}_\theta$ and $\boldsymbol{c}_\theta$ which define the posterior distributions. Inspired from recent progress in equivariant neural networks (Thomas et al., 2018; Fuchs et al., 2020; Satorras et al., 2021b; Guan et al., 2022), we model the interaction between the ligand molecule atoms and the protein atoms with a SE(3)-Equivariant GNN:

$$[\hat{\mathbf{x}}_0, \hat{\mathbf{v}}_0] = \phi_\theta(M_t, t, \mathcal{P}) = \phi_\theta([\mathbf{x}_t, \mathbf{v}_t], t, \mathcal{P}). \tag{6}$$

At the $l$-th layer, the atom hidden embedding $\mathbf{h}$ and coordinates $\mathbf{x}$ are updated alternately as follows:

$$\begin{aligned}
\mathbf{h}_i^{l+1} &= \mathbf{h}_i^l + \sum_{j \in \mathcal{V}, i \neq j} f_h(d_{ij}^l, \mathbf{h}_i^l, \mathbf{h}_j^l, \mathbf{e}_{ij}; \theta_h) \\
\mathbf{x}_i^{l+1} &= \mathbf{x}_i^l + \sum_{j \in \mathcal{V}, i \neq j} (\mathbf{x}_i^l - \mathbf{x}_j^l) f_x(d_{ij}^l, \mathbf{h}_i^{l+1}, \mathbf{h}_j^{l+1}, \mathbf{e}_{ij}; \theta_x) \cdot \mathbb{1}_{\text{mol}}
\end{aligned} \tag{7}$$

where $d_{ij} = \|\mathbf{x}_i - \mathbf{x}_j\|$ is the euclidean distance between two atoms $i$ and $j$ and $\mathbf{e}_{ij}$ is an additional feature indicating the connection is between protein atoms, ligand atoms or protein atom and ligand atom. $\mathbb{1}_{\text{mol}}$ is the ligand molecule mask since we do not want to update protein atom coordinates. The initial atom hidden embedding $\mathbf{h}^0$ is obtained by an embedding layer that encodes the atom information. The final atom hidden embedding $\mathbf{h}^L$ is fed into a multi-layer perceptron and a softmax function to obtain $\hat{\mathbf{v}}_0$. Since $\hat{\mathbf{x}}_0$ is rotation equivariant to $\mathbf{x}_t$ and it is easy to see $\mathbf{x}_{t-1}$ is rotation equivariant to $\mathbf{x}_0$ according to equation 4, we achieve the desired equivariance for Markov transition. The complete proof can be found in Appendix A.

### 3.5 TRAINING OBJECTIVE

The combination of $q$ and $p$ is a variational auto-encoder (Kingma and Welling, 2013). The model can be trained by optimizing the variational bound on negative log likelihood. For the atom coordinate loss, since $q(\mathbf{x}_{t-1}|\mathbf{x}_t, \mathbf{x}_0)$ and $p_\theta(\mathbf{x}_{t-1}|\mathbf{x}_t)$ are both Gaussian distributions, the KL-divergence can be written in closed form:

$$L_{t-1}^{(x)} = \frac{1}{2\sigma_t^2} \|\tilde{\boldsymbol{\mu}}_t(\mathbf{x}_t, \mathbf{x}_0) - \boldsymbol{\mu}_\theta([\mathbf{x}_t, \mathbf{v}_t], t, \mathcal{P})\|^2 + C = \gamma_t \|\mathbf{x}_0 - \hat{\mathbf{x}}_0\|^2 + C \tag{8}$$

where $\gamma_t = \frac{\bar{\alpha}_{t-1}\beta_t^2}{2\sigma_t^2(1-\bar{\alpha}_t)^2}$ and $C$ is a constant. In practice, training the model with an unweighted MSE loss (set $\gamma_t = 1$) could also achieve better performance as Ho et al. (2020) suggested. For the atom type loss, we can directly compute KL-divergence of categorical distributions as follows:

$$L_{t-1}^{(v)} = \sum_k \boldsymbol{c}(\mathbf{v}_t, \mathbf{v}_0)_k \log \frac{\boldsymbol{c}(\mathbf{v}_t, \mathbf{v}_0)_k}{\boldsymbol{c}(\mathbf{v}_t, \hat{\mathbf{v}}_0)_k}. \tag{9}$$

The final loss is a weighted sum of atom coordinate loss and atom type loss: $L = L_{t-1}^{(x)} + \lambda L_{t-1}^{(v)}$. We summarize the overall training and sampling procedure of TargetDiff in Appendix E.

## 3.6 AFFINITY RANKING AND PREDICTION AS UNSUPERVISED LEARNER

Generative models are unsupervised learners. However, in the area of target-aware molecule generation, nobody has established the connection between the generative model and binding affinity, which is an important indicator for evaluating generated molecules. Existing generative models can not (accurately) estimate the quality of generated molecules. Especially for models relying on autoregressive sampling, they have to assign an unnatural order when performing likelihood estimation (if possible) and cannot capture the global context as a whole.

We first establish the connection between unsupervised generative models and binding affinity ranking / prediction. Under our parameterization, the network predicts the denoised $[\hat{\mathbf{x}}_0, \hat{\mathbf{v}}_0]$. Given the protein-ligand complex, we can feed $\phi_\theta$ with $[\mathbf{x}_0, \mathbf{v}_0]$ while freezing the $\mathbf{x}$-update branch (i.e. only atom hidden embedding $\mathbf{h}$ is updated), and we could finally obtain $\mathbf{h}^L$ and $\hat{\mathbf{v}}_0$:

$$\mathbf{h}_i^{l+1} = \mathbf{h}_i^l + \sum_{j \in \mathcal{V}, i \neq j} f_h(d_{ij}^l, \mathbf{h}_i^l, \mathbf{h}_j^l, \mathbf{e}_{ij}; \theta_h)_{l=1\dots L-1} \quad \hat{\mathbf{v}}_0 = \texttt{softmax}(\text{MLP}(\mathbf{h}^L)). \quad (10)$$

Our assumption is that if the ligand molecule has a good binding affinity to protein, the flexibility of atom types should be low, which could be reflected in the entropy of $\hat{\mathbf{v}}_0$. Therefore, it can be used as a scoring function to help ranking, whose effectiveness is justified in the experiments. In addition, $\mathbf{h}^L$ also includes useful global information. We found the binding affinity ranking performance can be greatly improved by utilizing this feature with a simple linear transformation.

## 4 EXPERIMENTS

### 4.1 SETUP

**Data** We use CrossDocked2020 (Francoeur et al., 2020) to train and evaluate TargetDiff. Similar to Luo et al. (2021), we further refined the 22.5 million docked protein binding complexes by only selecting the poses with a low ($< 1\text{Å}$) and sequence identity less than 30%. In the end, we have 100,000 complexes for training and 100 novel complexes as references for testing.

**Baseline** For benchmarking, we compare with various baselines: **liGAN** (Ragoza et al., 2022), **AR** (Luo et al., 2021), **Pocket2Mol** (Peng et al., 2022), and **GraphBP** (Liu et al., 2022). **liGAN** is a 3D CNN-based method that generates 3D voxelized molecular images following a conditional VAE scheme. **AR**, **Pocket2Mol** and **GraphBP** are all GNN-based methods that generate 3D molecules by *sequentially* placing atoms into a protein binding pocket. We choose **AR** and **Pocket2Mol** as representative baselines with autoregressive sampling scheme because of their good empirical performance. All baselines are considered in Table 3 for a comprehensive comparison.

**TargetDiff** Our model contains 9 equivariant layers described in equation 7, where $f_h$ and $f_x$ are specifically implemented as graph attention layers with 16 attention heads and 128 hidden features. We first decide on the number of atoms for sampling by drawing a prior distribution estimated from training complexes with similar binding pocket sizes. After the model finishes the generative process, we then use OpenBabel (O'Boyle et al., 2011) to construct the molecule from individual atom coordinates as done in AR and liGAN. Please see Appendix F for the full details.

### 4.2 TARGET-AWARE MOLECULE GENERATION

We propose a comprehensive evaluation framework for target-aware molecule generation to justify the performance of our model and baselines from the following perspectives: **molecular structures**, **target binding affinity** and **molecular properties**.

**Molecular Structures** First, we plot the empirical distributions of all-atom distances and carbon-carbon bond distances in Figure 2, then compare them against the same empirical distributions for reference molecules. For overall atom distances, TargetDiff captures the overall distribution very well, while AR and Pocket2Mol has an over-representation for small atom distances. Due to its limited voxelized resolution, liGAN can only capture the overall shape but not specify modes. Similarly, different carbon-carbon bonds form two representative distance modes in reference molecular

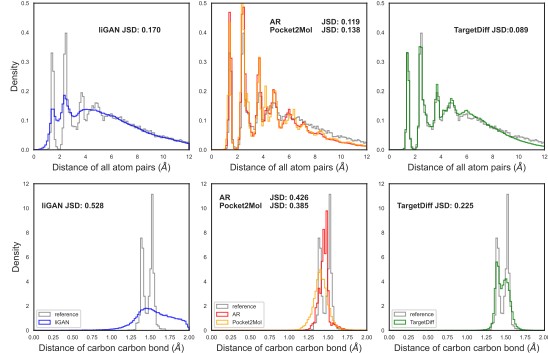

**Figure 2:** Comparing the distribution for distances of all-atom (top row) and carbon-carbon pairs (bottom row) for reference molecules in the test set (gray) and model generated molecules (color). Jensen-Shannon divergence (JSD) between two distributions is reported.

| Bond | liGAN | AR | Pocket2Mol | TargetDiff |
|------|-------|-------|-----------|-----------|
| C−C | 0.601 | 0.609 | 0.496 | **0.369** |
| C=C | 0.665 | 0.620 | 0.561 | **0.505** |
| C−N | 0.634 | 0.474 | 0.416 | **0.363** |
| C=N | 0.749 | 0.635 | 0.629 | **0.550** |
| C−O | 0.656 | 0.492 | 0.454 | **0.421** |
| C=O | 0.661 | 0.558 | 0.516 | **0.461** |
| C:C | 0.497 | 0.451 | 0.416 | **0.263** |
| C:N | 0.638 | 0.552 | 0.487 | **0.235** |

**Table 1:** Jensen-Shannon divergence between the distributions of bond distance for reference vs. generated molecules. "-", "=", and ":" represent single, double, and aromatic bonds, respectively. A lower value is better.

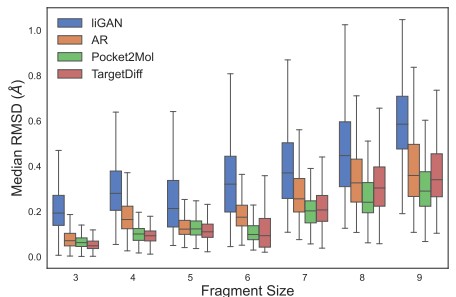

**Figure 3:** Median RMSD for rigid fragment before and after the force-field optimization.

| Ring Size | Ref. | liGAN | AR | Pocket2Mol | TargetDiff |
|-----------|------|-------|------|-----------|-----------|
| 3 | 1.7% | 28.1% | 29.9% | 0.1% | 0.0% |
| 4 | 0.0% | 15.7% | 0.0% | 0.0% | 2.8% |
| 5 | 30.2% | 29.8% | 16.0% | 16.4% | 30.8% |
| 6 | 67.4% | 22.7% | 51.2% | 80.4% | 50.7% |
| 7 | 0.7% | 2.6% | 1.7% | 2.6% | 12.1% |
| 8 | 0.0% | 0.8% | 0.7% | 0.3% | 2.7% |
| 9 | 0.0% | 0.3% | 0.5% | 0.1% | 0.9% |

**Table 2:** Percentage of different ring sizes for reference and model generated molecules.

structures. While we can still see the two modes in TargetDiff generated structures, only a single mode is observed for ones generated by liGAN, AR, and Pocket2Mol. In Table 1, we further evaluated how well different generated molecular structures capture the empirical distributions of bond distances in reference molecules, measured by Jensen-Shannon divergence (JSD) Lin (1991). We found that TargetDiff outperforms other methods with a clear margin across all major bond types.

Secondly, we investigate whether TargetDiff can generate rigid sub-structure / fragment in a consistent fashion (e.g., all carbons in a benzene ring are in the same plane). To measure such consistency, we optimize the generated structure with Merck Molecular Force Field (MMFF) Halgren (1996) and calculate the RMSD between pre-and pos- MMFF-optimized coordinates for different rigid fragments that do not contain any rotatable bonds. As shown in Figure 3, TargetDiff is able to generate more consistent rigid fragments. In a further analysis, we discover that liGAN and AR tend to generate a large amount of 3- and 4- member rings (Table 2). While TargetDiff shows a larger proportion of 7-member-ring, we believe this represents a limitation in the reconstruction algorithm and could be an interesting future direction to replace such post-hoc operation with bond generation.

These results suggest that TargetDiff can produce more realistic molecular structures throughout the process compared to existing baselines and therefore perceive a more accurate 3D representation of the molecular dynamics leading to better protein binders.

**Target Binding Affinity** Figure 4 shows the median Vina energy (computed by AutoDock Vina (Eberhardt et al., 2021)) of all generated molecules for each binding pocket. Based on the Vina energy, generated molecules from TargetDiff show the best binding affinity in **57%** of the targets, while the ones from liGAN, AR and Pocket2Mol are only best for 4%, 13% and 26% of all targets. In terms of high-affinity binder, we find that on average **58.1%** of the TargetDiff molecules show better binding affinity than the reference molecule, which is clearly better than other baselines (See

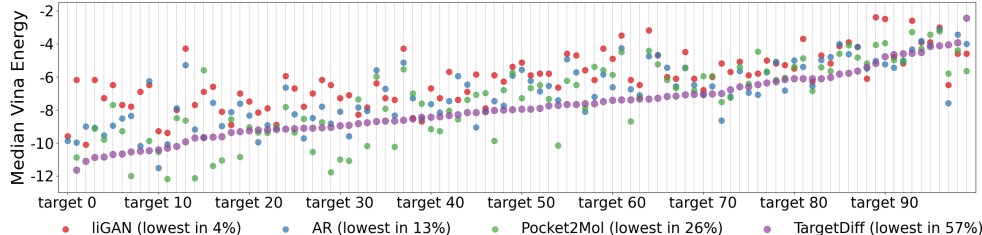

**Figure 4:** Median Vina energy for different generated molecules (liGAN vs. AR vs. TargetDiff) across 100 testing binding targets. Binding targets are sorted by the median Vina energy of TargetDiff generated molecules. Lower Vina energy means a higher estimated binding affinity.

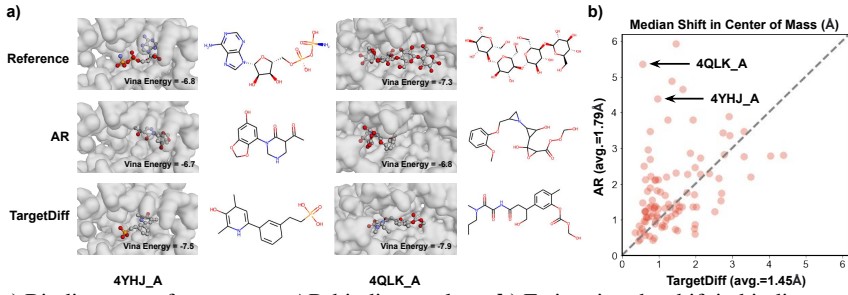

**Figure 5: a)** Binding poses for two poor-AR-binding pockets. **b)** Estimating the shift in binding poses between generated and reference molecules by calculating the distance between their center of mass (CoM). Each point represents the median CoM distance for one target.

Table 3). We further compute Vina Score and Vina Min in Table 3, where the Vina score function is directly computed or locally optimized without re-docking. They directly reflect the quality of model generated 3D molecules and similarly, our model outperforms all other baselines.

To better understand the differences in generated molecules, we sample a generated molecule from each model for two pockets where TargetDiff outperforms AR. As shown in Figure 5a, while TargetDiff can generate molecules occupying the entire pocket, AR is only able to generate a molecule that covers part of the space and potentially loses its specificity for the desired target and cause off-target effects. Let us consider the AR-generated molecules for 4QLK_A. Despite having a similar number of atoms as the TargetDiff molecule (27 vs. 29), the frontier network in AR keeps placing molecules deep inside the pocket instead of considering the global structure, and trying to cover the entire binding pocket results in poor binding affinity. To further quantify such effects, we measure the distance between the center of mass (CoM) for reference molecules and the CoM for generated molecules. As shown in Figure 5b, the sequential generation nature of AR results in a larger shift in CoM (1.79Å vs. 1.45Å) and presents sub-optimal binding poses with poorer binding affinity.

**Molecular Properties** Besides binding affinity, we further investigate other molecular properties for generated molecules, including drug likeliness QED (Bickerton et al., 2012), synthesizability SA (Ertl and Schuffenhauer, 2009; You et al., 2018), and diversity computed as the average pairwise

| Metric / Model | Vina Score (↓) | | Vina Min (↓) | | Vina Dock (↓) | | High Affinity (↑) | | QED (↑) | | SA (↑) | | Diversity (↑) | |
|---|---|---|---|---|---|---|---|---|---|---|---|---|---|---|
| | Avg. | Med. | Avg. | Med. | Avg. | Med. | Avg. | Med. | Avg. | Med. | Avg. | Med. | Avg. | Med. |
| liGAN * | - | - | - | - | -6.33 | -6.20 | 21.1% | 11.1% | 0.39 | 0.39 | 0.59 | 0.57 | 0.66 | 0.67 |
| GraphBP * | - | - | - | - | -4.80 | -4.70 | 14.2% | 6.7% | 0.43 | 0.45 | 0.49 | 0.48 | **0.79** | **0.78** |
| AR | **-5.75** | -5.64 | -6.18 | -5.88 | -6.75 | -6.62 | 37.9% | 31.0% | 0.51 | 0.50 | 0.63 | 0.63 | 0.70 | 0.70 |
| Pocket2Mol | -5.14 | -4.70 | -6.42 | -5.82 | -7.15 | -6.79 | 48.4% | 51.0% | **0.56** | **0.57** | **0.74** | **0.75** | 0.69 | 0.71 |
| TargetDiff | -5.47 | **-6.30** | **-6.64** | **-6.83** | **-7.80** | **-7.91** | **58.1%** | **59.1%** | 0.48 | 0.48 | 0.58 | 0.58 | 0.72 | 0.71 |
| Reference | -6.36 | -6.46 | -6.71 | -6.49 | -7.45 | -7.26 | - | - | 0.48 | 0.47 | 0.73 | 0.74 | - | - |

**Table 3:** Summary of different properties of reference molecules and molecules generated by our model and other baselines. For liGAN and GraphBP, AutoDock Vina could not parse some generated atom types and thus we use QVina (Alhossary et al., 2015) to perform docking. See additional evaluation results in Appendix G.

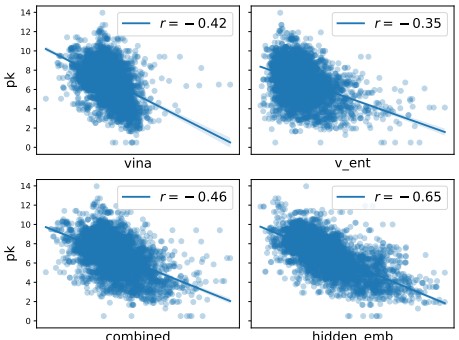

**Figure 6:** Binding affinity ranking results on CrossDocked2020. Spearman's rank correlation coefficients between different indicators and experimentally measured binding affinity are shown.

**Table 4:** Binding affinity prediction results on PDBbind v2020. EGNN augmented with our unsupervised features achieves best results on all four metrics.

| Metric / Model | RMSE ↓ | Pearson ↑ | Spearman ↑ | MAE ↓ |
|---|---|---|---|---|
| TransCPI | 1.741 | 0.576 | 0.540 | 1.404 |
| MONN | 1.438 | 0.624 | 0.589 | 1.143 |
| IGN | 1.433 | 0.698 | 0.641 | 1.169 |
| HOLOPROT | 1.546 | 0.602 | 0.571 | 1.208 |
| STAMP-DPI | 1.658 | 0.545 | 0.411 | 1.325 |
| EGNN | 1.445 | 0.648 | 0.598 | 1.141 |
| EGNN + ours | 1.374 | 0.680 | 0.637 | 1.118 |

Tanimoto distances (Bajusz et al., 2015; Tanimoto, 1958). As shown in Table 3, TargetDiff can generate more high-affinity binders compared to liGAN, AR, and GraphBP while maintaining similar other 2D metrics. The metrics TargetDiff does fall behind Pocket2Mol are the QED and SA scores. However, we put less emphasis on them because in the context of drug discovery, QED and SA are used as rough filter and would be fine as long as they are in a reasonable range. Therefore, they might not be the metrics we want to optimize against. We believe future investigation around prediction on bonds and fragment-based (instead of atom-based) generation could lead to improvement.

### 4.3 Binding Affinity Ranking and Prediction

To justify that our model can serve as an unsupervised learner to improve the binding affinity ranking and prediction, we first check Spearman's rank correlation on CrossDocked2020. AutoDock Vina score (*i.e.* vina) and negative log-transformed experimentally measured binding affinity pK are provided along with the dataset. As shown in Figure 6, we found: (1) The entropy of denoised atom type $\hat{\mathbf{v}}_0$ (*i.e.* v_ent) has a reasonable correlation with pK, indicating unsupervised learning can provide a certain degree of information for binding affinity ranking. (2) The entropy score provides some complementary information to traditional chemical / physical-based score function like Vina, since the combination of them (*i.e.* combined) achieves better correlation. (3) When provided with labeled data, the final hidden embedding $\mathbf{h}^L$ (*i.e.* hidden_emb) with a simple linear transformation could improve the correlation to a large extent.

We further demonstrate that our unsupervised learned features could improve supervised affinity prediction on PDBBind v2020 dataset (Liu et al., 2015). We perform a more difficult time split as Stärk et al. (2022) in which the test set consists of structures deposited after 2019, and the training and validation set consist of earlier structures. We augment EGNN (Satorras et al., 2021b) with the unsupervised features $\mathbf{h}^L$ provided by our model, and compare it with two state-of-the-art sequence-based models TransCPI (Chen et al., 2020) and MONN (Li et al., 2020), one complex model IGN (Jiang et al., 2021), two structure-based model HOLOPROT (Somnath et al., 2021) and STAMP-DPI (Wang et al., 2022) and finally the base EGNN model. As shown in Table 4, the augmented EGNN clearly improves the vanilla EGNN and achieves the best results among baselines.

## 5 Conclusion

This paper proposes TargetDiff, a 3D equivariant diffusion model for target-aware molecule generation and enhancing binding affinity prediction. In terms of future work, it would be interesting to incorporate bond generation as part of the diffusion process such that we can skip the bond inference algorithm. In addition to bond inference, another interesting future direction would be incorporating some of the techniques in fragment-based molecule generation Podda et al. (2020) and generating molecules with common and more synthesizable molecular sub-structures.

**Reproducibility Statements**  The model implementation, experimental data and model check-points can be found here: `https://github.com/guanjq/targetdiff`

**Acknowledgement**  We thank all the reviewers for their feedbacks through out the review cycles of the manuscript. This work was supported by National Key R&D Program of China No. 2021YFF1201600, U.S. National Science Foundation under grant no. 2019897 and U.S. Department of Energy award DE-SC0018420.

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

# A   PROOF OF SE(3)-EQUIVARIANCE OF GENERATIVE MARKOV TRANSITION

One crucial property the model needs to satisfy is that rotating or translating the protein-ligand complex $(\mathcal{M}, \mathcal{P})$ will not change the estimated likelihood $p_\theta(\mathcal{M}|\mathcal{P})$. Leveraging the conclusion from Köhler et al. (2020); Xu et al. (2022), it requires the initial density of our generative process $p(M_T|\mathcal{P})$ is SE(3)-invariant and the Markov transition $p_\theta(M_{t-1}|M_t, \mathcal{P})$ is SE(3)-equivariant. Since atom types are always invariant to SE(3)-transformation during the generative process, we only need to consider the atom coordinates. More concretely, the model needs to satisfy:

$$p(\mathbf{x}_T, \mathbf{x}_P) = p(T_g(\mathbf{x}_T, \mathbf{x}_P))$$
$$p(\mathbf{x}_{t-1}|\mathbf{x}_t, \mathbf{x}_P) = p(T_g(\mathbf{x}_{t-1})|T_g(\mathbf{x}_t, \mathbf{x}_P))$$

where $T_g$ is the group of SE(3)-transformation, $\mathbf{x}_t$ and $\mathbf{x}_P$ denote the atom coordinates of ligand molecule and protein separately. $T_g(\mathbf{x})$ can also be written explicitly as $T_g(\mathbf{x}) = \boldsymbol{R}\mathbf{x} + \boldsymbol{b}$, where $\boldsymbol{R} \in \mathbb{R}^{3\times 3}$ is the rotation matrix and $\boldsymbol{b} \in \mathbb{R}^3$ is the translation vector.

In Sec. 3.4, we provide a way to implement SE(3)-equivariance of the generative Markov transition. In this section, we will prove the SE(3)-equivariance of our design.

Recall the equation 7, we update the atom hidden embedding $\mathbf{h}$ and coordinates $\mathbf{x}$ alternately as follows:

$$\mathbf{h}_i^{l+1} = \mathbf{h}_i^l + \sum_{j \in \mathcal{V}, i \neq j} f_h(d_{ij}^l, \mathbf{h}_i^l, \mathbf{h}_j^l, \mathbf{e}_{ij}; \theta_h)$$
$$\mathbf{x}_i^{l+1} = \mathbf{x}_i^l + \sum_{j \in \mathcal{V}, i \neq j} (\mathbf{x}_i^l - \mathbf{x}_j^l) f_x(d_{ij}^l, \mathbf{h}_i^{l+1}, \mathbf{h}_j^{l+1}, \mathbf{e}_{ij}; \theta_x) \cdot \mathbb{1}_{\text{mol}}$$

First, it is easy to see $d_{ij}$ does not change with the 3D roto-translation $T_g$:

$$\hat{d}_{ij}^2 = \|T_g(\mathbf{x}_i) - T_g(\mathbf{x}_j)\|^2 = \|(\boldsymbol{R}\mathbf{x}_i + \boldsymbol{b}) - (\boldsymbol{R}\mathbf{x}_j + \boldsymbol{b})\|^2 = \|\boldsymbol{R}\mathbf{x}_i - \boldsymbol{R}\mathbf{x}_j\|^2$$
$$= (\mathbf{x}_i - \mathbf{x}_j)^\top \boldsymbol{R}^\top \boldsymbol{R}(\mathbf{x}_i - \mathbf{x}_j) = (\mathbf{x}_i - \mathbf{x}_j)^\top \mathbf{I}(\mathbf{x}_i - \mathbf{x}_j) = \|\mathbf{x}_i - \mathbf{x}_j\|^2 = d_{ij}^2$$

Since $\mathbf{h}_i, \mathbf{h}_j, \mathbf{e}_{ij}$ are initially obtained from atom and edge features, which are invariant to SE(3)-transformation, we have $\mathbf{h}_i^l$ is SE(3)-invariant for any $l = 1, \ldots, L$.

Then, we can prove that $\mathbf{x}$ updated from equation 7 is SE(3)-equivariant as follows:

$$\phi_\theta(T(\mathbf{x}^l)) = T(\mathbf{x}_i^l) + \sum_{j \in \mathcal{V}, i \neq j} (T(\mathbf{x}_i^l) - T(\mathbf{x}_j^l)) f_x(d_{ij}^l, \mathbf{h}_i^{l+1}, \mathbf{h}_j^{l+1}, \mathbf{e}_{ij}; \theta_x) \cdot \mathbb{1}_{\text{mol}}$$
$$= \boldsymbol{R}\mathbf{x}_i^l + \boldsymbol{b} + \sum_{j \in \mathcal{V}, i \neq j} \boldsymbol{R}(\mathbf{x}_i^l - \mathbf{x}_j^l) f_x(d_{ij}^l, \mathbf{h}_i^{l+1}, \mathbf{h}_j^{l+1}, \mathbf{e}_{ij}; \theta_x) \cdot \mathbb{1}_{\text{mol}}$$
$$= \boldsymbol{R}\left( \mathbf{x}_i^l + \sum_{j \in \mathcal{V}, i \neq j} \boldsymbol{R}(\mathbf{x}_i^l - \mathbf{x}_j^l) f_x(d_{ij}^l, \mathbf{h}_i^{l+1}, \mathbf{h}_j^{l+1}, \mathbf{e}_{ij}; \theta_x) \cdot \mathbb{1}_{\text{mol}} \right) + \boldsymbol{b} \tag{11}$$
$$= \boldsymbol{R}\mathbf{x}_i^{l+1} + \boldsymbol{b}$$
$$= T(\phi_\theta(\mathbf{x}^l))$$

Under our parameterization, the neural network predicts $[\hat{\mathbf{x}}_0, \hat{\mathbf{v}}_0]$. By stacking $L$ such equivariant layers together, we can draw the conclusion that the output of neural network $\hat{\mathbf{x}}_0$ is SE(3)-equivariant w.r.t the input $\mathbf{x}_t$. Finally, we can obtain the mean of posterior $\hat{\mathbf{x}}_{t-1}$ from equation 4: $\hat{\mathbf{x}}_{t-1} = \frac{\sqrt{\bar{\alpha}_{t-1}}\beta_t}{1 - \bar{\alpha}_t}\hat{\mathbf{x}}_0 + \frac{\sqrt{\alpha_t}(1 - \bar{\alpha}_{t-1})}{1 - \bar{\alpha}_t}\mathbf{x}_t$. The last thing the model needs to satisfy is that $\hat{\mathbf{x}}_{t-1}$ is SE(3)-equivariant w.r.t $\mathbf{x}_t$. However, we can see the translation vector will be changed under this formula:

$$\boldsymbol{\mu}_\theta(T(\mathbf{x}_t), t) = \frac{\sqrt{\bar{\alpha}_{t-1}}\beta_t}{1 - \bar{\alpha}_t}T(\hat{\mathbf{x}}_0) + \frac{\sqrt{\alpha_t}(1 - \bar{\alpha}_{t-1})}{1 - \bar{\alpha}_t}T(\mathbf{x}_t)$$
$$= \frac{\sqrt{\bar{\alpha}_{t-1}}\beta_t}{1 - \bar{\alpha}_t}\boldsymbol{R}(\hat{\mathbf{x}}_0) + \frac{\sqrt{\alpha_t}(1 - \bar{\alpha}_{t-1})}{1 - \bar{\alpha}_t}\boldsymbol{R}(\mathbf{x}_t) + \tilde{\boldsymbol{b}} \tag{12}$$

where $\tilde{\boldsymbol{b}} = \left( \frac{\sqrt{\bar{\alpha}_{t-1}}\beta_t}{1-\bar{\alpha}_t} + \frac{\sqrt{\alpha_t}(1-\bar{\alpha}_{t-1})}{1-\bar{\alpha}_t} \right) \boldsymbol{b}$

As the Sec. 3.4 discussed, we can move CoM of the protein atoms to zero once to achieve translation invariance in the whole generative process, which is same to how we achieve the invariant initial density. Thus, we only need to consider rotation equivariance in the Markov transition, which is straightforward to see that it can be achieved from equation 12 when $\boldsymbol{b}$ is ignored: $\boldsymbol{\mu}_\theta(\boldsymbol{R}(\mathbf{x}_t), t) = \boldsymbol{R}(\boldsymbol{\mu}_\theta(\mathbf{x}_t, t))$.

## B    ANALYSIS OF INVARIANT INITIAL DENSITY

We assume when the timestep $T$ of the diffusion process is sufficiently large, $q(\mathbf{x}_T|\mathbf{x}_P)$ would be a Gaussian distribution whose mean is the center of protein and standard deviation is one, i.e. $q(\mathbf{x}_T|\mathbf{x}_P) \sim \mathcal{N}(C_P \otimes \mathbf{1}_{N_P}, I_{3 \cdot N_P})$, where $C_P = \frac{1}{N_P} \sum \mathbf{x}_P$, $\otimes$ denotes the kronecker product, $I_k$ denotes the $k \times k$ identity matrix and $\mathbf{1}_k$ denotes the $k$-dimensional vector filled with one.

To achieve the SE(3)-invariant initial density, we move the center of protein to zero, i.e. $C_P = 0$. One can also define the initial density on other CoM-free systems such as the ligand or complex CoM-free system. We choose protein CoM-free system here since only one step of shifting center operation is needed at the beginning of generative or diffusion process (protein context is a fixed input). Formally, it can be considered as a linear transformation: $\hat{\mathbf{x}}_P = Q\mathbf{x}_P$, where $Q = I_3 \otimes (I_N - \frac{1}{N}\mathbf{1}_N\mathbf{1}_N^T)$. It has several benefits in simplifying the formula: In the diffusion process, $q(\hat{\mathbf{x}}_T|\hat{\mathbf{x}}_P)$ would be a standard Gaussian when $T$ is sufficiently large; Accordingly, in the generative process, we can sample $\hat{\mathbf{x}}_T$ from $p(\hat{\mathbf{x}}_T|\hat{\mathbf{x}}_P)$, which is also a standard Gaussian distribution.

For evaluating a complex position $(\mathbf{x}_T, \mathbf{x}_P)$, we can firstly translate the complex to achieve zero CoM on protein positions, which can also be considered as a linear transformation:

$$(\hat{\mathbf{x}}_T, \hat{\mathbf{x}}_P) = Q(\mathbf{x}_T, \mathbf{x}_P), \quad \text{where} \quad Q = I_3 \otimes \begin{pmatrix} I_M & -\frac{1}{N}\mathbf{1}_M\mathbf{1}_N^T \\ \mathbf{0} & I_N - \frac{1}{N}\mathbf{1}_N\mathbf{1}_N^T \end{pmatrix} \tag{13}$$

Then, we can evaluate the density $p(\hat{\mathbf{x}}_T|\hat{\mathbf{x}}_P)$ with the standard normal distribution. We denote $\hat{p}$ as the density function on this protein zero CoM subspace: $\hat{p}(\mathbf{x}_T|\mathbf{x}_P) = p(Q(\mathbf{x}_T, \mathbf{x}_P))$

It can be seen that for any rigid transformation $T_g(\mathbf{x}) = \boldsymbol{R}\mathbf{x} + \boldsymbol{b}$, we have $Q \cdot T_g(\mathbf{x}_T, \mathbf{x}_P) = Q \cdot \boldsymbol{R}(\mathbf{x}_T, \mathbf{x}_P)$. Since $Q$ is a symmetric projection operator and rotation matrix $\boldsymbol{R}$ is a orthogonal matrix, we have $\|Q \cdot R\mathbf{x}\|^2 = \|\mathbf{x}\|^2$. Given $p$ is an isotropic normal distribution, we can easily have $\hat{p}(T_g(\mathbf{x}_T, \mathbf{x}_P)) = \hat{p}(\mathbf{x}_T, \mathbf{x}_P)$, which means an SE(3)-invariant density.

## C    PROOF OF INVARIANT LIKELIHOOD

In Sec. 3.4, we argue that an invariant initial density composed with an equivariant transition function will result in an invariant distribution. In this section, we will provide the proof of it.

The two conditions to guarantee an invariant likelihood $p_\theta(M_0|\mathcal{P})$ are as follows:

$$p(\mathbf{x}_T, \mathbf{x}_P) = p(T_g(\mathbf{x}_T, \mathbf{x}_P)) \qquad \text{(①  Invariant Prior)}$$
$$p(\mathbf{x}_{t-1}|\mathbf{x}_t, \mathbf{x}_P) = p(T_g(\mathbf{x}_{t-1})|T_g(\mathbf{x}_t, \mathbf{x}_P)) \qquad \text{(②  Equivariant Transition)}$$

We can obtain the conclusion as follows:

$$p_\theta(T_g(\mathbf{x}_0, \mathbf{x}_P)) = \int p(T_g(\mathbf{x}_T, \mathbf{x}_P)) \sum_{t=1}^{T} p_\theta(T_g(\mathbf{x}_{t-1})|T_g(\mathbf{x}_t, \mathbf{x}_P)))$$

$$= \int p(\mathbf{x}_T, \mathbf{x}_P) \sum_{t=1}^{T} p_\theta(T_g(\mathbf{x}_{t-1})|T_g(\mathbf{x}_t, \mathbf{x}_P))) \qquad \leftarrow \text{Apply ①}$$

$$= \int p(\mathbf{x}_T, \mathbf{x}_P) \sum_{t=1}^{T} p_\theta(\mathbf{x}_{t-1}|\mathbf{x}_t, \mathbf{x}_P) \qquad \leftarrow \text{Apply ②}$$

$$= p_\theta(\mathbf{x}_0, \mathbf{x}_P)$$

# D  DERIVATION OF ATOM TYPES DIFFUSION PROCESS

According to Bayes theorem, we have:

$$
\begin{aligned}
q(\mathbf{v}_{t-1}|\mathbf{v}_t, \mathbf{v}_0) &= \frac{q(\mathbf{v}_t|\mathbf{v}_{t-1}, \mathbf{v}_0)q(\mathbf{v}_{t-1}|\mathbf{v}_0)}{\sum_{\mathbf{v}_{t-1}} q(\mathbf{v}_t|\mathbf{v}_{t-1}, \mathbf{v}_0)q(\mathbf{v}_{t-1}|\mathbf{v}_0)} \\
&= \frac{q(\mathbf{v}_t|\mathbf{v}_{t-1})q(\mathbf{v}_{t-1}|\mathbf{v}_0)}{\sum_{\mathbf{v}_{t-1}} q(\mathbf{v}_t|\mathbf{v}_{t-1})q(\mathbf{v}_{t-1}|\mathbf{v}_0)}
\end{aligned}
\tag{14}
$$

According to Eq. 2 and 3, $q(\mathbf{v}_t|\mathbf{v}_{t-1})$ and $q(\mathbf{v}_{t-1}|\mathbf{v}_0)$ can be calculated as:

$$
\begin{aligned}
q(\mathbf{v}_t|\mathbf{v}_{t-1}) &= \mathcal{C}(\mathbf{v}_t|\alpha_t\mathbf{v}_{t-1} + (1 - \alpha_t)/K) \\
q(\mathbf{v}_{t-1}|\mathbf{v}_0) &= \mathcal{C}(\mathbf{v}_{t-1}|\bar{\alpha}_{t-1}\mathbf{v}_0 + (1 - \bar{\alpha}_{t-1})/K)
\end{aligned}
\tag{15}
$$

Note that when computing $\mathcal{C}(\mathbf{v}_t|\alpha_t\mathbf{v}_{t-1} + (1 - \alpha_t)/K)$, the value is $\alpha_t + (1 - \alpha_t)/K$ if $\mathbf{v}_t = \mathbf{v}_{t-1}$ and $(1 - \alpha_t)/K$ otherwise, which leads to symmetry of this function Hoogeboom et al. (2021), i.e., $\mathcal{C}(\mathbf{v}_t|\alpha_t\mathbf{v}_{t-1} + (1 - \alpha_t)/K) = \mathcal{C}(\mathbf{v}_{t-1}|\alpha_t\mathbf{v}_t + (1 - \alpha_t)/K)$.

Let $\boldsymbol{c}^\star(\mathbf{v}_t, \mathbf{v}_0)$ denotes the numerator of Eq. 14. Then it can be computed as:

$$
\begin{aligned}
\boldsymbol{c}^\star(\mathbf{v}_t, \mathbf{v}_0) &= q(\mathbf{v}_t|\mathbf{v}_{t-1})q(\mathbf{v}_{t-1}|\mathbf{v}_0) \\
&= [\alpha_t\mathbf{v}_t + (1 - \alpha_t)/K] \odot [\bar{\alpha}_{t-1}\mathbf{v}_0 + (1 - \bar{\alpha}_{t-1})/K]
\end{aligned}
\tag{16}
$$

and therefore the posterior of atom types is derived as:

$$
q(\mathbf{v}_{t-1}|\mathbf{v}_t, \mathbf{v}_0) = \mathcal{C}(\mathbf{v}_{t-1}|\tilde{\boldsymbol{c}}_t(\mathbf{v}_t, \mathbf{v}_0))
\tag{17}
$$

where $\tilde{\boldsymbol{c}}_t(\mathbf{v}_t, \mathbf{v}_0) = \boldsymbol{c}^\star / \sum_{k=1}^K c_k^\star$

# E  OVERALL TRAINING AND SAMPLING PROCEDURES

In this section, we summarize the overall training and sampling procedures of TargetDiff as Algorithm 1 and Algorithm 2 respectively.

---

**Algorithm 1** Training Procedure of TargetDiff

---

**Input:** Protein-ligand binding dataset $\{\mathcal{P}, \mathcal{M}\}_{i=1}^N$, neural network $\phi_\theta$
 1: **while** $\phi_\theta$ not converge **do**
 2:    Sample diffusion time $t \in \mathcal{U}(0, \ldots, T)$
 3:    Move the complex to make CoM of protein atoms zero
 4:    Perturb $\mathbf{x}_0$ to obtain $\mathbf{x}_t$: $\mathbf{x}_t = \sqrt{\bar{\alpha}_t}\mathbf{x}_0 + (1 - \bar{\alpha}_t)\epsilon$, where $\epsilon \in \mathcal{N}(0, \boldsymbol{I})$
 5:    Perturb $\mathbf{v}_0$ to obtain $\mathbf{v}_t$:
      $\log \boldsymbol{c} = \log\left(\bar{\alpha}_t\mathbf{v}_0 + (1 - \bar{\alpha}_t)/K\right)$
      $\mathbf{v}_t = \texttt{one\_hot}(\arg\max_i[g_i + \log c_i])$, where $g \sim \text{Gumbel}(0, 1)$
 6:    Predict $[\hat{\mathbf{x}}_0, \hat{\mathbf{v}}_0]$ from $[\mathbf{x}_t, \mathbf{v}_t]$ with $\phi_\theta$: $[\hat{\mathbf{x}}_0, \hat{\mathbf{v}}_0] = \phi_\theta([\mathbf{x}_t, \mathbf{v}_t], t, \mathcal{P})$
 7:    Compute the posterior atom types $\boldsymbol{c}(\mathbf{v}_t, \mathbf{v}_0)$ and $\boldsymbol{c}(\mathbf{v}_t, \hat{\mathbf{v}}_0)$ according to equation 4
 8:    Compute the unweighted MSE loss on atom coordinates and the KL loss on posterior atom types: $L = \|\mathbf{x}_0 - \hat{\mathbf{x}}_0\|^2 + \alpha\, \text{KL}(\boldsymbol{c}(\mathbf{v}_t, \mathbf{v}_0) \parallel \boldsymbol{c}(\mathbf{v}_t, \hat{\mathbf{v}}_0))$
 9:    Update $\theta$ by minimizing $L$
10: **end while**

---

# F  EXPERIMENT DETAILS

## F.1  FEATURIZATION

At the $l$-th layer, we dynamically construct the protein-ligand complex as a $k$-nearest neighbors (knn) graph based on known protein atom coordinates and current ligand atom coordinates, which

---

**Algorithm 2** Sampling Procedure of TargetDiff

---

**Input:** The protein binding site $\mathcal{P}$, the learned model $\phi_\theta$.
**Output:** Generated ligand molecule $\mathcal{M}$ that binds to the protein pocket.
 1: Sample the number of atoms in $\mathcal{M}$ based on a prior distribution conditioned on the pocket size
 2: Move CoM of protein atoms to zero
 3: Sample initial molecular atom coordinates $\mathbf{x}_T$ and atom types $\mathbf{v}_T$:
        $\mathbf{x}_T \in \mathcal{N}(0, \boldsymbol{I})$
        $\mathbf{v}_T = \texttt{one\_hot}(\arg\max_i g_i)$, where $g \sim \text{Gumbel}(0, 1)$
 4: **for** $t$ in $T, T-1, \ldots, 1$ **do**
 5:     Predict $[\hat{\mathbf{x}}_0, \hat{\mathbf{v}}_0]$ from $[\mathbf{x}_t, \mathbf{v}_t]$ with $\phi_\theta$: $[\hat{\mathbf{x}}_0, \hat{\mathbf{v}}_0] = \phi_\theta([\mathbf{x}_t, \mathbf{v}_t], t, \mathcal{P})$
 6:     Sample $\mathbf{x}_{t-1}$ from the posterior $p_\theta(\mathbf{x}_{t-1}|\mathbf{x}_t, \hat{\mathbf{x}}_0)$ according to equation 4
 7:     Sample $\mathbf{v}_{t-1}$ from the posterior $p_\theta(\mathbf{v}_{t-1}|\mathbf{v}_t, \hat{\mathbf{v}}_0)$ according to equation 4
 8: **end for**

---

is the output of the $l-1$-th layer. We choose $k = 32$ in our experiments. The protein atom features include chemical elements, amino acid types and whether the atoms are backbone atoms. The ligand atom types are one-hot vectors consisting of the chemical element types and aromatic information. The edge features are the outer products of distance embedding and bond types, where we expand the distance with radial basis functions located at 20 centers between 0 Å and 10 Å and the bond type is a 4-dim one-hot vector indicating the connection is between protein atoms, ligand atoms, protein-ligand atoms or ligand-protein atoms.

## F.2 MODEL PARAMETERIZATION

Our model consists of 9 equivariant layers as equation 7 shows, and each layer is a Transformer with `hidden_dim=128` and `n_heads=16`. The key/value embedding and attention scores are generated through a 2-layer MLP with LayerNorm and ReLU activation. We choose to use a sigmoid $\beta$ schedule with $\beta_1 = \texttt{1e-7}$ and $\beta_T = \texttt{2e-3}$ for atom coordinates, and a cosine $\beta$ schedule suggested in Nichol and Dhariwal (2021) with $s = 0.01$ for atom types. We set the number of diffusion steps as 1000.

## F.3 TRAINING DETAILS

The model is trained via gradient descent method Adam Kingma and Ba (2014) with `init_learning_rate=0.001`, `betas=(0.95, 0.999)`, `batch_size=4` and `clip_gradient_norm=8`. We multiply a factor $\alpha = 100$ on the atom type loss to balance the scales of two losses. During the training phase, we add a small Gaussian noise with a standard deviation of 0.1 to protein atom coordinates as data augmentation. We also schedule to decay the learning rate exponentially with a factor of 0.6 and a minimum learning rate of 1e-6. The learning rate is decayed if there is no improvement for the validation loss in 10 consecutive evaluations. The evaluation is performed for every 2000 training steps.

We trained our model on one NVIDIA GeForce GTX 3090 GPU, and it could converge within 24 hours and 200k steps.

## F.4 DETERMINE THE NUMBER OF LIGAND ATOMS

**Pocket Size Estimation**   We compute the top 10 farthest pairwise distances of protein atoms, and select the median of it as the pocket size for robustness.

**Prior Distribution of Number of Ligand Atoms**   We compute 10 quantiles of training pocket sizes and estimate the prior distribution of number of ligand atoms for each bin. Specifically, we take the histogram of number of atoms in the training set as the prior distribution. The relationship between estimated prior distribution and actual training testing number of atom distribution is shown in Figure S1. We can see that the number of ligand atoms has a clear positive correlation with pocket sizes and the prior distribution estimated from the training set can also be generalized to the test set.

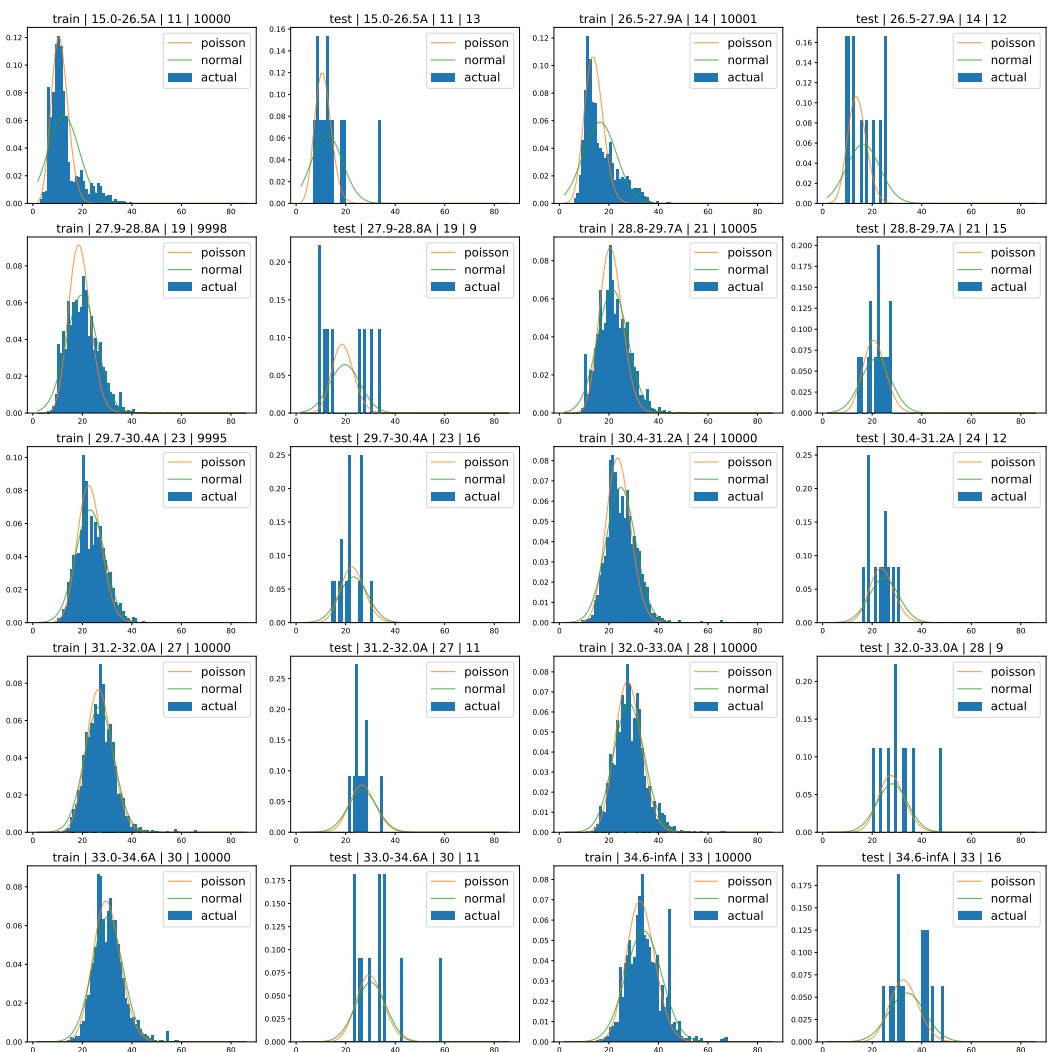

**Figure S1:** Prior distributions of the number of ligand atoms. For each prior distribution within a specific range of pocket size, *train/test | pocket size range | median number of ligand atoms | number of data points within current bin* are shown as titles.

During the training phase, we provide the model with the number of atoms of reference molecules since we use them to perform training. During the generation phase, we do not have to require the generated molecules has the same number of atoms as the reference molecule, and the numbers are randomly sampled from these prior distributions computed based on training data.

## G ADDITIONAL EVALUATION RESULTS

In the main text, we provided the evaluation results in Table. 3 where the Vina score is computed with AutoDock Vina (Eberhardt et al., 2021). Here, we provided the evaluation results in Table S1 where Vina scores are computed with QVina (Alhossary et al., 2015), a faster but less accurate docking tool (following what AR and Pocket2Mol used). We can observe a similar trend that molecules generated by TargetDiff could achieve SOTA binding affinity.

Upon further investigation, we also discover a strong negative correlation between SA score and molecular size as shown in Figure S2 (Pearson R=$-0.56$, $p \leq 10^{-80}$), and the difference in SA score between these model generated molecules could be the artifact of their size differences.

| | Metric
Model | High Affinity (↑) | | QED (↑) | | SA (↑) | | Diversity (↑) | |
|---|---|---|---|---|---|---|---|---|---|
| | | Avg. | Med. | Avg. | Med. | Avg. | Med. | Avg. | Med. |
| liGAN | | 21.1% | 11.1% | 0.39 | 0.39 | 0.59 | 0.57 | 0.66 | 0.67 |
| AR | | 33.7% | 24.2% | 0.51 | 0.50 | 0.63 | 0.63 | 0.70 | 0.70 |
| GraphBP | | 14.2% | 6.7% | 0.43 | 0.45 | 0.49 | 0.48 | 0.79 | 0.78 |
| Pocket2Mol | | 49.8% | 52.0% | 0.56 | 0.57 | 0.74 | 0.75 | 0.69 | 0.71 |
| TargetDiff | | 51.3% | 50.0% | 0.48 | 0.48 | 0.58 | 0.58 | 0.72 | 0.71 |
| Reference | | - | - | 0.48 | 0.47 | 0.73 | 0.74 | - | - |

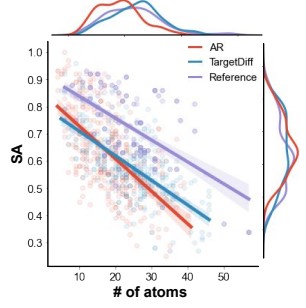

**Table S1:** Summary of different properties of reference molecules and molecules generated by our model and other baselines.

**Figure S2:** A scatter plot compares the SA score and the number of atoms for a given molecule.

## H    SAMPLING TIME ANALYSIS

One major advantage of TargetDiff over auto-regressive based model such AR is that TargetDiff scales better against the size of the molecules. While AR is required to run additional steps to generate larger molecules, the diffusion model can operate on additional atoms in a parallel fashion without sacrificing a lot of inference time.

To better demonstrate such effect, we randomly select 5 binding pockets as targets and record the time spent in generating 100 molecules for each pocket using both autoregressive models (including AR, Pocket2Mol and GraphBP) and TargetDiff. Since these models have different numbers of parameters and sampling schemes, we first compare the inference time ratio against generating a 10-atom molecule for these models, instead of simply comparing their wall time. As shown in Figure S3, as we start to generate larger and larger molecules, the wall time for AR grows almost linearly along with the molecule size, while the wall time for TargetDiff stays relatively flat.

In terms of wall clock time, AR, Pocket2Mol and GraphBP use 7785s, 2544s and 105s for generating 100 valid molecules on average separately, and it takes 3428s on average for TargetDiff. GraphBP has the fastest sampling time but the quality of generated molecules is lower than other models (See Table 3). TargetDiff has the moderate sampling efficiency compared to AR and Pocket2Mol.

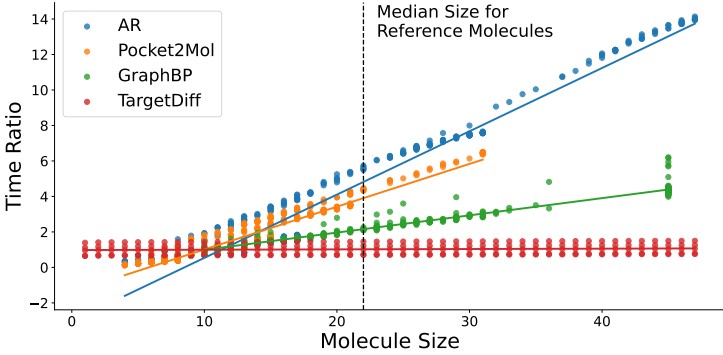

**Figure S3:** Inference time growth as a function of molecule size for AR, Pocket2Mol, GraphBP and TargetDiff.

## I    MORE EXAMPLES OF GENERATED RESULTS

Please see next page.

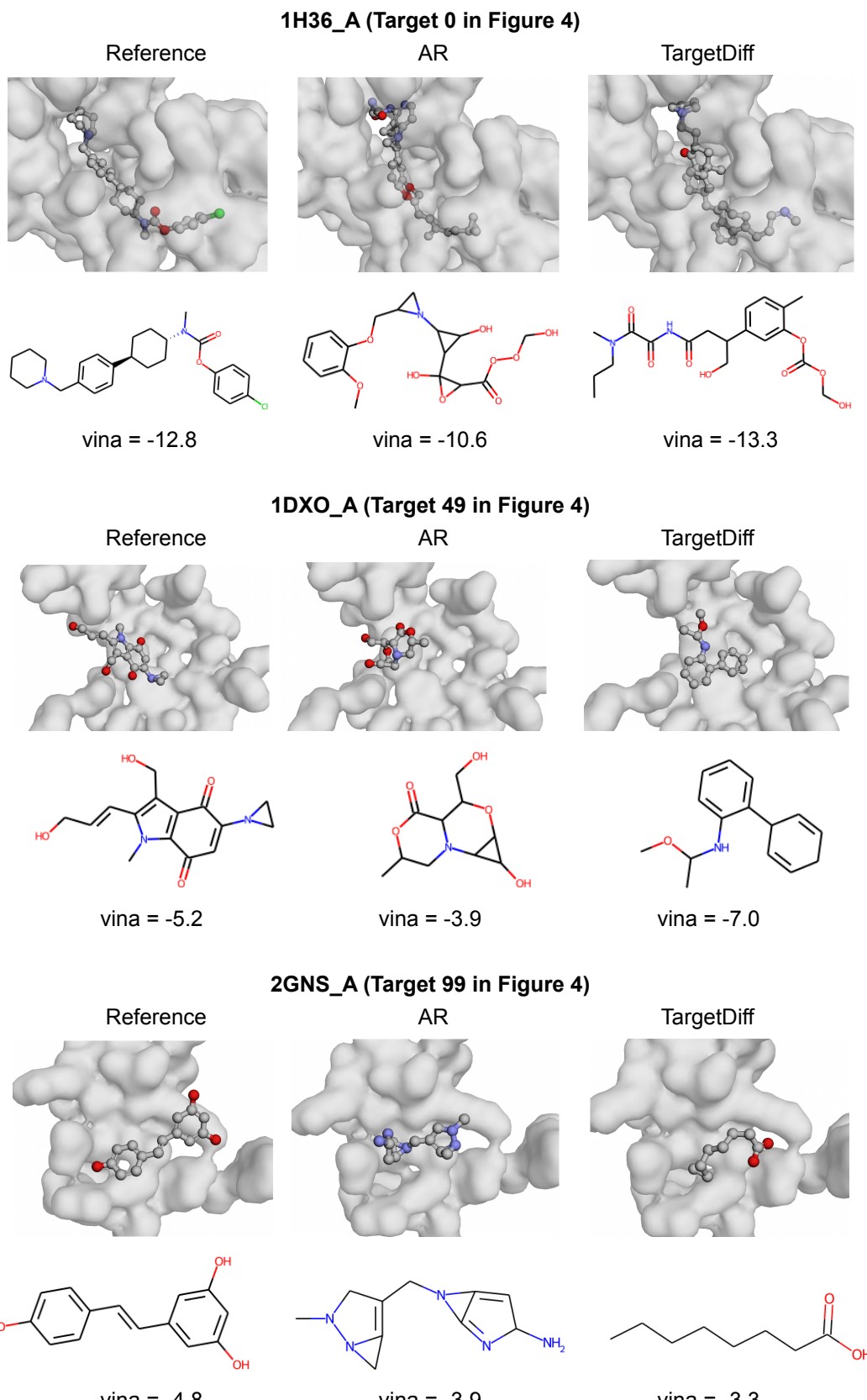

**Figure S4:** More examples of binding poses for generated molecules. To present of fair overview of the model performance, we specifically select the best (1H36_A), median (1DXO_A), and worst (2GNS_A) targets shown in Figure 4 for visualization along with the generated molecules and calculated Vina energy (kcal/mol).

## J    EXAMPLES WHERE AR OUTPERFORMS TARGETDIFF

Please see followings for two binding targets where AR outperforms TargetDiff in terms of Vina estimated binding affinity.

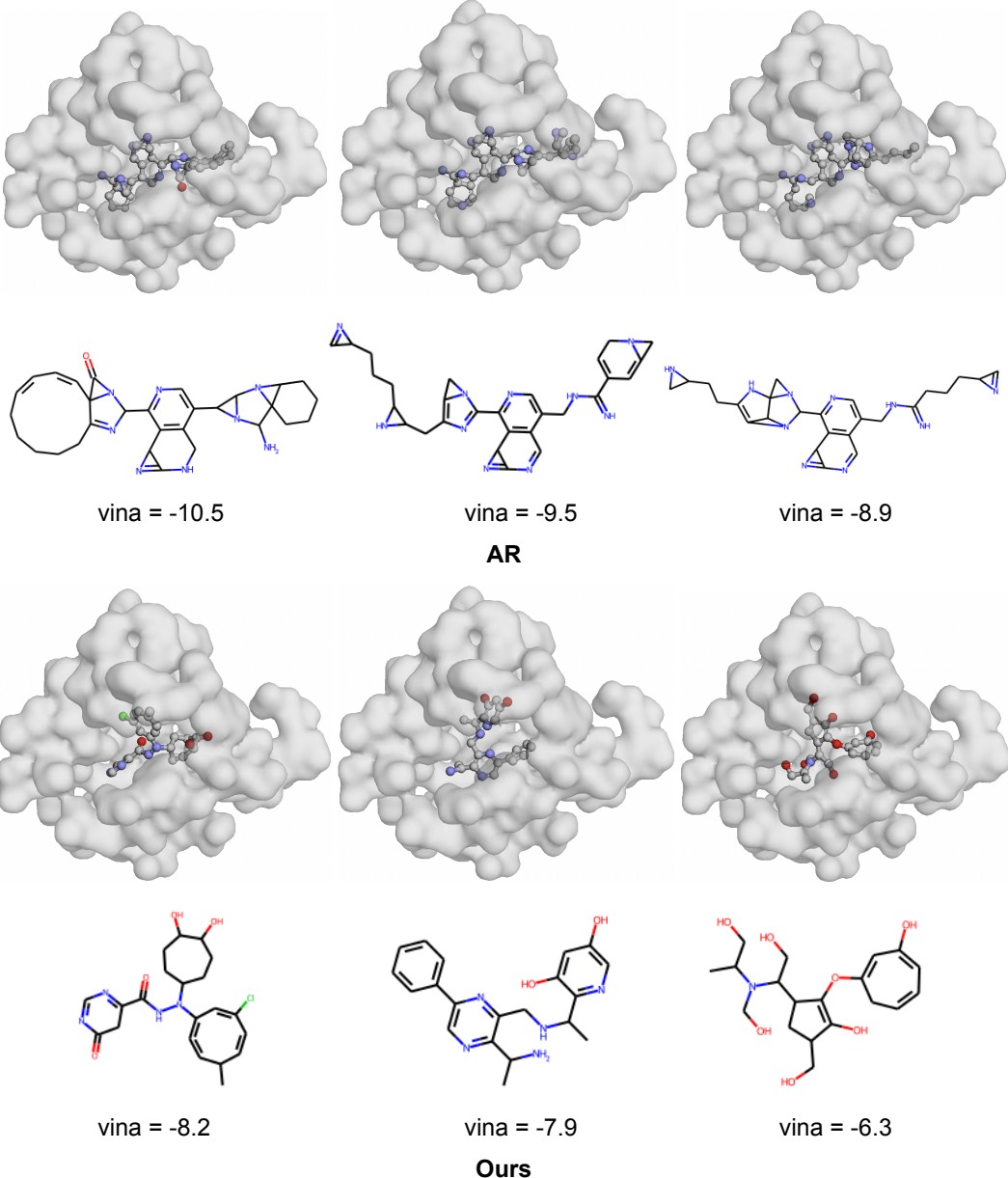

**Figure S5:** Example binding pocket 1 (4KCQ_A, target 33 in Figure 4) where AR outperforms TargetDiff in terms of Vina estimated binding affinity. Three examples of AR generated molecules are shown in the top half, and three examples from TargetDiff are shown in the bottom half.

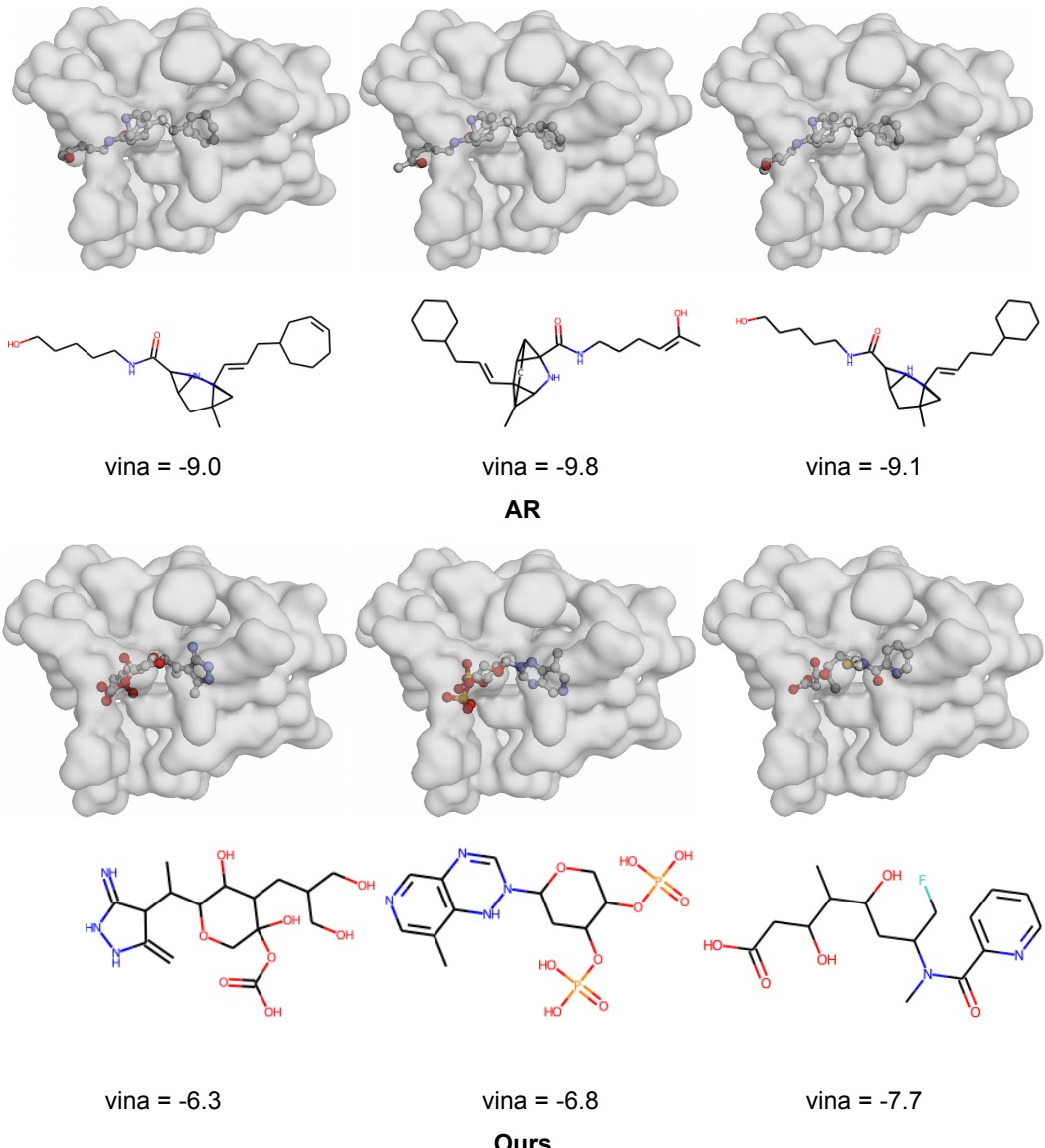

**Figure S6:** Example binding pocket 1 (1R1H_A, target 61 in Figure 4) where AR outperforms TargetDiff in terms of Vina estimated binding affinity. Three examples of AR generated molecules are shown in the top half, and three examples from TargetDiff are shown in the bottom half.

