# OpenReview forum: "3D Equivariant Diffusion for Target-Aware Molecule Generation and Affinity Prediction"
_ICLR.cc/2023/Conference — ICLR 2023 poster_

### Official Review · Reviewer_3Z2H · 2022-10-17

**Confidence:** 4
**Correctness:** 3
**Technical Novelty And Significance:** 2
**Empirical Novelty And Significance:** 2
**Recommendation:** 5

**Clarity, Quality, Novelty And Reproducibility:**

Clarity & Quality: This paper is well-written and easy to follow. The model and experiment seem correct.

Novelty: This method seems to be a combination of diffusion models, equivariant networks, and structure-based molecular generation tasks.

Reproducibility: The paper provides the details for Reproducibility, but some are missed, refer to the main review.


**Strength And Weaknesses:**

This paper tackles the problem of structure-based drug design using diffusion models and SE(3)-equivariant networks. However, there are some concerns about the method and the experiments:
- This method seems to be a combination of diffusion models, equivariant networks, and structure-based molecular generation tasks. Compared with [1], which combines diffusion models and equivariant networks, the innovation of this work is not obvious. It seems the only difference is the atom information in the protein target is used. The authors can highlight more about its innovation compared with previous works.
- The number of atoms is sampled by drawing a prior distribution estimated from training complexes with similar binding pocket sizes. As determining the number of atoms in advance is crucial to the generation of the diffusion model, I wonder if the authors can show the difference between the sampled number of atoms and the number of atoms of reference molecules. What will happen if the error is too large to cover the reference molecules?
- The details of defining "similar binding pocket sizes" is not provided in the paper.
- The detail of the data split is not clear. To avoid leakage, it is important to exclude similar pockets in training data.
- It seems the test set is from the CrossDocked2020, not the experimental crystal protein-ligand complex.
- Vina will first "re-docking" the molecules and then calculate the docking score. Since re-docking can largely change the molecular conformation and binding pose, I wonder if the authors have considered this problem when evaluating the binding affinity?
    - I would suggest providing the Vina score without re-docking, to demonstrate the end-to-end 3D generation performance, rather than post-fixed by docking tools.
- In many results, only AR is used as baselines. And From table 3, there are several baselines better than AR. I think these baselines should be included in most tables/figures.
- In Table 3, TargetDiff seems only marginally better (and worse in some metrics) than other baselines, especially compared with Pocket2Mol.
- The quality of generated molecules. From Figure 5, S2, and S3, It seems TargetDiff tends to generate the molecule with large flexibility, which is unstable in the real world.
- Did you use any tools, like rdkit/openbabel, to post-fix and filter the generated molecules?
- This paper compares the efficiency of TargetDiff and AR in the appendix. Since GraphBP and Pocket2Mol in Table 3 are also auto-regressive generative models especially Pocket2Mol adopts a more efficient sampling method than MCMC sampling in AR, I wonder how efficient TargetDiff is when compared with Pocket2Mol and GraphBP.
- Can you show the wall clock time, rather than the ratio, in the efficiency comparison?
- The model architecture is not clear. In Sec. 3.4, it says "we model the interaction between the ligand molecule atoms and the protein atoms with a SE(3)-Equivariant GNN". But in Sec. 4.1, it says "where fh and fx are specifically implemented as transformers with 16 attention heads and 128 hidden features." I am confused about the statements, is the model a GNN or a transformer?


[1] Emiel Hoogeboom, Victor Garcia Satorras, Clément Vignac, and Max Welling. Equivariant diffusion for molecule generation in 3d. In International Conference on Machine Learning, pages 8867–8887. PMLR, 2022.


**Summary Of The Paper:**

This paper tackles the problem of generating molecules for a specific protein target. With diffusion models and SE(3)-equivariant networks, this paper learns a joint generative process of both continuous atom coordinates and categorical atom types. Specifically, this paper defines a diffusion process for the continuous atom coordinates and discrete atom types, gradually adding noise, and learns the joint generative process for atom types and coordinates of molecules.
Besides, this paper evaluates the quality of generated molecules based on the features outputted by the model, which can be used as a scoring function for ranking or binding affinity prediction.


**Summary Of The Review:**

Given the incremental novelty and the marginal improvement in empirical experiments, I recommend the rejection.

---

> ### Author Response · Authors · 2022-11-14
> **Response to Reviewer 3Z2H (Part 1 of 2)**
>
> We thank the reviewer for the feedback and suggestions. Please see below for our responses to the comments.
>
> **Q1: “This method seems to be a combination of diffusion models, equivariant networks, and structure-based molecular generation tasks. … The authors can highlight more about its innovation compared with previous works”**
>
> A1: 1) While there are existing generative models for structured-based drug design, most of them are derived from the autoregressive sampling scheme. In this work, we instead propose an effective generative model with non-autoregressive sampling scheme which aligns training and sampling process. 2) While the techniques in our work have been described in other context, we are the first to effectively apply these techniques on the target-aware molecule generation task, which are quite different from other applications. 3) Last but not the least, we show that our model can be served as an unsupervised binding affinity predictor with comprehensive experiments, which has never been described in prior literature. Such a predictor can be used for rank-based screening of generated molecules and can also improve supervised binding affinity prediction.
>
> **Q2: “I wonder if the authors can show the difference between the sampled number of atoms and the number of atoms of reference molecules. What will happen if the error is too large to cover the reference molecules?”**
>
> A2: Great questions! We have included the figures about distributions of number of atoms in **Appendix Figure S1**.
> During the training phase, we provide the model with the number of atoms of reference molecules since we use them to perform training.
> During the generation phase, we don’t have to require the generated molecules has the same number of atoms as the reference molecule, and the numbers are randomly sampled from a prior distribution computed based on training data.
>
> **Q3: The details of defining "similar binding pocket sizes" is not provided in the paper.**
>
> A3: Thank you for pointing this out! We compute the top 10 farthest pairwise distances of protein atoms, and select the median of it as the pocket size for robustness. We have updated **Appendix F.4** to include the details.
>
> **Q4: The detail of the data split is not clear. To avoid leakage, it is important to exclude similar pockets in training data.**
>
> A4: As pointed out in the text, “In the end, we have 100,000 complexes for training and 100 novel complexes as references for testing.” and there are no pocket in the test set that is similar to the ones in training set. This split also aligns with previous baselines.
>
> **Q5: It seems the test set is from the CrossDocked2020, not the experimental crystal protein-ligand complex.**
>
> A5: For such a generation task, we could not do more things (like better evaluation metrics)  with experimental crystal protein-ligand complex as the test set, excepting that it may set a stronger “reference” baseline, which is probably not guaranteed since crystal protein-ligand complex may also have bad binding affinity. Again, our evaluation is consistent with all prior literature.
>
> **Q6: Vina will first "re-docking" the molecules and then calculate the docking score. Since re-docking can largely change the molecular conformation and binding pose, I wonder if the authors have considered this problem when evaluating the binding affinity?**
>
> A6: Thank you for your suggestion! We totally agree that computing Vina score with re-docking can not reflect the real generation performance of various models, and we have provided Vina Score, Vina Min and Vina Dock computed with AutoDock Vina [1] in Table 3, and observed a similar trend that molecules generated by our model could achieve SOTA binding affinity. Prior results with QVina [2] (which is used in AR and Pocket2Mol) have been moved to **Appendix G** instead.
>
> **Q7: In many results, only AR is used as baselines. And From table 3, there are several baselines better than AR. I think these baselines should be included in most tables/figures.**
>
> A7: Taking the feedback from Reviewer uzYP, **we have compared with the strongest baseline Pocket2Mol thoroughly (Table 1, 2, 3, Figure 2, 3, 4, and relevant text are all updated).**

---

> > ### Author Response · Authors · 2022-11-14
> > **Response to Reviewer 3Z2H (Part 2 of 2)**
> >
> > **Q8: In Table 3, TargetDiff seems only marginally better (and worse in some metrics) than other baselines, especially compared with Pocket2Mol.**
> >
> > A8: From the updated tables and figures, it can be seen our model outperforms existing baselines (including Pocket2Mol) in terms of atomic distance distribution, bond distance distribution, and Autodock Vina binding affinity with a clear margin, and on-par with Pocket2Mol in rigid body rmsd and ring distribution. Note that these are results obtained without vector-neuron based networks as Pocket2Mol used, which indicates the superiority of our diffusion modeling, which captures the global information better compared to the prior art.
> >
> > We put less emphasis on the QED and SA scores because 1. They are metrics to evaluate generated 2D molecular graphs. Pocket2Mol has advantages in these metrics because it predicts bonds. It would be an interesting future direction to extend TargetDiff to modeling bonds. 2. In the context of drug discovery, QED and SA are used as rough filters and would be fine as long as they are in a reasonable range. Therefore, they might not be the metrics we want to optimize against. On the other hand, people cares more about the value of binding affinity to the protein targets.
> >
> > **Q9: The quality of generated molecules. From Figure 5, S2, and S3, It seems TargetDiff tends to generate the molecule with large flexibility, which is unstable in the real world.**
> >
> > A9: In practice, the chemists expect the generated molecules has plane substructure (like aromatic rings) and also has some degree of flexibility to avoid over-complanation. Except the last molecule in Figure S2 has too large flexibility (this is the one visualization example where our model performs worse compared to other baselines), we believe the generated molecules observe similar size and flexibility with the reference molecules as well.
> >
> > **Q10: Did you use any tools, like rdkit/openbabel, to post-fix and filter the generated molecules?**
> > A10: We only use openbabel to add bonds. In all experiments, only Vina Min, Vina Dock (previously QVina Dock) post-fix the generated molecules.
> >
> > **Q11: I wonder how efficient TargetDiff is when compared with Pocket2Mol and GraphBP. Can you show the wall clock time, rather than the ratio, in the efficiency comparison?**
> > A11: Thank you for the suggestions! Please see the updated **Appendix H** for details. We found that as we start to generate larger and larger molecules, the wall time for all autoregressive models (AR, Pocket2Mol and GraphBP)  grows almost linearly along with the molecule size, while the wall time for TargetDiff stays relatively flat.  In terms of wall clock time, AR, Pocket2Mol and GraphBP use 7785s, 2544s and 105s for generating 100 valid molecules on average separately, and it takes 3428s on average for TargetDiff. GraphBP has the fastest sampling time but the quality of generated molecules is lower than other models (As Table 3 shows). TargetDiff has a moderate sampling efficiency compared to AR and Pocket2Mol.
> >
> > **Q12: The model architecture is not clear. In Sec. 3.4, it says "we model the interaction between the ligand molecule atoms and the protein atoms with a SE(3)-Equivariant GNN". But in Sec. 4.1, it says "where fh and fx are specifically implemented as transformers with 16 attention heads and 128 hidden features." I am confused about the statements, is the model a GNN or a transformer?**
> >
> > A12: GNN with attention mechanism which is similar to the one used in transformer. We have fixed the description in Sec. 4.1.
> >
> > **References**
> >
> > [1] Eberhardt, J., Santos-Martins, D., Tillack, A. F., & Forli, S. (2021). AutoDock Vina 1.2. 0: New docking methods, expanded force field, and python bindings. Journal of Chemical Information and Modeling, 61(8), 3891-3898.
> >
> > [2] Alhossary, A., Handoko, S. D., Mu, Y., & Kwoh, C. K. (2015). Fast, accurate, and reliable molecular docking with QuickVina 2. Bioinformatics, 31(13), 2214-2216.

---

> > > ### Comment · Reviewer_3Z2H · 2022-11-18
> > > **Thank you for the response!**
> > >
> > > Thank authors so much for the detailed response, many of my concerns/questions are addressed.
> > > However, for question 1, I still cannot understand clearly about the contribution, especially compared with [1].
> > >
> > > [1] Emiel Hoogeboom, Victor Garcia Satorras, Clément Vignac, and Max Welling. Equivariant diffusion for molecule generation in 3d. In International Conference on Machine Learning, pages 8867–8887. PMLR, 2022.

---

> > > > ### Author Response · Authors · 2022-11-19
> > > > **Thank you for the feedback and additional clarification on the contribution**
> > > >
> > > > Thank you for the response and involving the discussion! Below are additional clarifications on our contributions, especially comparing with EDM.
> > > >
> > > > 1) First, we want to emphasize the **task** we want to solve is important and substantially different from EDM [Hoogeboom et al., 2022]. In EDM, only experiments on **unbounded molecular generation** are performed (on QM9/GEOM-Drugs dataset), and metrics such as validity, uniqueness, and stability are used to evaluate, which is hard to be directly applied in real scenarios and utilized by biologists. One would have no clues on detecting biological activities and functions of these generated molecules. In contrast, TargetDiff works in a more realistic setting where proteins of interest are considered in the model. **We are the first work trying to solve this task with a non-autoregressive SE(3)-equivariant model and achieved SOTA results.** Diffusion model is an ideal fit for this task, which enables us design neural networks directly in 3D space (without voxelization) and avoid potential problems existing in autoregressive models. We believe offering a solution to a key biochemical problem using a novel class of generative model is an important step toward this direction and could give some inspiration to future work, which definitely falls under the scope of ICLR -- biological applications.
> > > >
> > > > 2) In addition, recent work for structure-based drug design (AR, Pocket2Mol, GraphBP) have been evaluated with SA, QED and Vina docking score, which may have bias as we illustrate in Appendix G and as you said (thank you again for pointing out the problem in Vina docking!). We proposed a **new and comprehensive** set of evaluation criteria from the perspectives of molecular structures, target binding affinity and molecular properties, and achieved SOTA results.
> > > >
> > > > 3) Although TargetDiff shares some commons with EDM, we made substantial changes to make it work as the first diffusion model in the 3D conditional generation scenario for drug design. For example, in preliminary experiments we found EGNN worked less well in this setting, and thus we chose to use a more powerful SE(3)-equivariant GNN with attention mechanism; we show how to achieve **equivariance in the conditional setting** (Appendix A-C); Importantly, the choice of formulating atom type as a categorical distribution and parameterize NN to predict denoised atom positions and atom types allow us to construct an effective **unsupervised binding affinity predictor**! It has never been described in prior literature and explicitly differs from EDM, which could not build such a predictor with continuous atom type representations and recovered noise as the NN's output.
> > > >
> > > > 4) Finally, we do recognize EDM as a very relevant work in our context, and we have discussed them in Related Work section.
> > > >
> > > > We hope these clarifications could address your concern about the contribution. Please let us know if there are things that we could do to help improve your initial rating. Thank you!

---

> > > > > ### Comment · Reviewer_3Z2H · 2022-11-25
> > > > > **Thank you for the further response**
> > > > >
> > > > > 1. The reviewer agrees the task is essential. However, compared with EDM, which already was non-autoregressive, the contribution of targetdiff seems minor, since the main difference in the diffusion process is the involvement of $\mathcal{P}$.
> > > > >
> > > > > 2. The authors claim that targetdiff is the first SE(3)-equivariant non-autoregressive pocket-based 3D generation. However, the baseline Ligan (voxel based) is a non-autoregressive generation work. And the voxel-based method is naturally rotationally equivariant.
> > > > >
> > > > > 3. From eq(7), I believe the SE(3)-equivariance is still based on EGNN, which leverages the equivariant delta coordinates.
> > > > >
> > > > > 4. To summarize, at the model/architecture level, I think the contribution of the targetdiff is minor (a pocket condition $\mathcal{P}$ and attention in the model), compared with EDM.
> > > > >
> > > > > 5. There are several other contributions, like the "binding affinity predictor", and more evaluation metrics. However, if the contributions of the main model are significant, these additional contributions will be a good "icing on the cake".

---

> > > > > > ### Author Response · Authors · 2022-11-25
> > > > > > **Thank you for the further feedback**
> > > > > >
> > > > > > We thank the reviewer for the additional feedback.
> > > > > >
> > > > > > We want to point out that liGAN is indeed a non-autoregressive generation work since it's based on VAE, but it is **not SE(3)-equivariant** because it uses 3D CNN to encode protein and ligand. The standard 3D convolution is only equivariant to translation but not rotation. We refer [1] to the reviewer, which has some discussion about the equivariance of 3D CNN. In addition, we also mentioned that such voxelization operation may lead to poor scalability issues in the main text. We believe there is no doubt for our claim that "targetdiff is the first SE(3)-equivariant non-autoregressive pocket-based 3D generation model".
> > > > > >
> > > > > > We agree that the empirical significance of our work overweights the technical significance, but leveraging a proper technique to solve an essential scientific problem is also an important step.
> > > > > >
> > > > > > [1] Worrall, D., & Brostow, G. (2018). Cubenet: Equivariance to 3d rotation and translation. In Proceedings of the European Conference on Computer Vision (ECCV) (pp. 567-584).

---

> > > > > > > ### Comment · Reviewer_3Z2H · 2022-12-07
> > > > > > > **Thank you**
> > > > > > >
> > > > > > > Thank the author for the detailed responses and the revisions in the paper.
> > > > > > > I have no more questions, and will increase the score due to the effort made by the author during the Discussion stage.
> > > > > > > However, as the performance improvement seems insignificant, I still feel slightly negative about the paper.

---

### Official Review · Reviewer_uzYP · 2022-10-23

**Confidence:** 4
**Correctness:** 3
**Technical Novelty And Significance:** 2
**Empirical Novelty And Significance:** 2
**Recommendation:** 5

**Clarity, Quality, Novelty And Reproducibility:**

**Clarity:** There are no major concerns with respect to the clarity. I was able to follow the paper.

**Quality:** In principle, the paper is well executed with a lot of experiments. However, I believe some strong baselines have been missed (see weaknesses above). If the paper claims to be generally superior to autoregressive models as one of its motivations, I would expect the authors to thoroughly compare to the very best and most comparable autoregressive approaches (such as Pocket2Mol).

**Novelty:** The methodological novelty is small. The paper is a straightforward combination of existing techniques (Gaussian and discrete diffusion models to model 3D atom coordinates and features in a protein ligand complex using equivariant neural networks). I think the observation that the entropy of the atom feature distribution correlates with binding affinity is interesting, but this is just one small experiment. Probably there could be done more in that direction. I am wondering whether even better unsupervised binding affinity predictors could be constructed.

**Reproducibility:** I do not have any concerns regarding the reproducibility. The paper uses public datasets and seems to provide sufficient training details, such that the experiments could in principle be re-run. The required compute resources are also very modest.

**Strength And Weaknesses:**

**Strengths:**

- The particular combination of using a mixed continuous-discrete diffusion model, together with an equivariant architecture, to model atom coordinates and features in a protein ligand complex seems to be new (although it's a straight-forward combination, see below).

- The work's quantitative results look okay and outperform the two chosen baselines liGAN and AR, although I am not sure the baseline comparisons are sufficient.

- It is interesting that the entropy of the distribution of generated atom features correlates with binding affinity.

**Weaknesses:**

- Methodologically, the method is a straight-forward combination of existing techniques. The Gaussian and categorical diffusion processes used are well established. The equivariant neural network architectures seem to be standard. And jointly modeling the ligand-protein complex with a generative model is also not new. Merely this exact combination of techniques seems to be new.

- The paper makes claims that the equivariant diffusion model approach is better than autoregressive techniques in its introduction and motivation. Yet, the method is outperformed by Pocket2Mol -- an autoregressive method -- with regards to the molecular properties (Table 3, approximately similar performance on affinity and diversity, significantly worse on SA and QED). In the other experiments, the Pocket2Mol baseline is not even considered. However, I believe it would be very appropriate to compare to this work more rigorously, as it also uses modern equivariant architectures, considers similar molecule generation tasks with the same data, and also has code available.

- The model only predicts 3D atom coordinates and atom types and features, but no bonds.

**Summary Of The Paper:**

The paper tackles target-aware small molecule generation. Specifically, small synthetic ligand molecules are generated using a deep generative model, such that the molecules best fit into the binding site of a larger protein. To this end, the paper jointly models the ligand and the protein using a generative diffusion model, conditioning the generation of the ligand on the protein. Technically, in contrast to some previous papers, the work explicitly models 3D atom coordinates and also uses appropriate equivariant neural networks. Diffusion processes over both atom coordinates and also atom types and features are used. The paper also shows the entropy of the generated atom feature distribution can serve as an approximate measure for binding affinity. Empirically, the paper compares mostly to two simple baselines, a 3D voxel-based molecule generative model and an autoregressive model, and outperforms them on various molecule generation tasks.

**Summary Of The Review:**

In summary, this is an okay paper with thorough experiments. Nevertheless, I do not think it meets the bar at this point, mainly due to two concerns: (a) the methodological novelty is not very significant, as discussed above. (b) This would be okay, if the experimental results would be very strong. However, it seems the paper is somewhat overclaiming in that regard, as it mostly compares to only two baselines that do not seem to represent the strongest existing methods. In particular, the method is motivated as an approach that can generally outperform autoregressive methods. Yet, only a simple autoregressive model ("AR") seems to be considered in most experiments, but not methods such as Pocket2Mol. In conclusion, I do not think the paper is ready yet for publication. I would recommend the authors to more thoroughly compare to baselines. Furthermore, it could be interesting to investigate even further how the learnt features can be used for property prediction, beyond just using the entropy as a measure for binding affinity.

---

> ### Author Response · Authors · 2022-11-14
> **Response to Reviewer uzYP**
>
> We thank the reviewer for the feedback and suggestions. Please see below for our responses to the comments.
>
> **Q1: The method is a straightforward combination of existing techniques. The methodological novelty is not very significant.**
>
> A1: 1) While there are existing generative models for structured-based drug design, most of them are derived from the autoregressive sampling scheme. In this work, we instead propose an effective generative model with non-autoregressive sampling scheme which aligns training and sampling process. 2) While the techniques in our work have been described in other context, we are the first to effectively apply these techniques on the target-aware molecule generation task, which are quite different from other applications. 3) Last but not the least, we show that our model can be served as an unsupervised binding affinity predictor with comprehensive experiments, which has never been described in prior literature. Such a predictor can be used for rank-based screening of generated molecules and can also improve supervised binding affinity prediction.
>
> **Q2: The paper mostly compares to only two baselines that do not seem to represent the strongest existing methods.  Only a simple autoregressive model ("AR") seems to be considered in most experiments, but not methods such as Pocket2Mol.**
>
> A2: Thank you for the suggestion! We initially only compared with AR because it has the most similar modeling (predict atom type and atom coordinates) and network architecture (GNN with attention) to our model. **We have updated our manuscript and compared our model with Pocket2Mol thoroughly (Table 1, 2, 3, Figure 2, 3, 4, and relevant text are all updated).** To reflect the comments of Reviewer 3Z2H, we also add Vina Score, Vina Min in Table 3, which are scores computed by AutoDock Vina without re-docking. The High Affinity ratio is also updated based on the AutoDock Vina docking score, which is a more accurate estimation than QVina we used before. It can be seen our model outperforms existing baselines (including Pocket2Mol) in terms of atomic distance distribution, bond distance distribution, and Autodock Vina binding affinity with a clear margin. Note that these are results obtained without vector-neuron based networks as Pocket2Mol used, which indicates the superiority of our diffusion modeling, which captures the global information better compared to the prior art.
>
> We put less emphasis on the QED and SA scores because 1. They are metrics to evaluate generated 2D molecular graphs. Pocket2Mol has advantages in these metrics because it predicts bonds. It would be an interesting future direction to extend TargetDiff to modeling bonds. (As you also have pointed, thanks for the suggestion!) 2. In the context of drug discovery, QED and SA are used as rough filters and would be fine as long as they are in a reasonable range. Therefore, they might not be the metrics we want to optimize against. On the other hand, people cares more about the value of binding affinity to the protein targets.

---

> > ### Author Response · Authors · 2022-11-19
> > **Any further comments from reviewer uzYP?**
> >
> > Dear Reviewer,
> >
> > Thanks for your valuable suggestions which help us improve our manuscript! We have updated our manuscript and compared our model with Pocket2Mol thoroughly (Table 1, 2, 3, Figure 2, 3, 4, and relevant text are all updated) as you suggested. As the reviewer-AC discussion period approaches, we want to check in again and see if there are additional concerns we can address for you to consider raising the score? Thanks!

---

> > > ### Comment · Reviewer_uzYP · 2022-11-22
> > > **Thank you for your response**
> > >
> > > I would like to thank the authors for their reply and for the additional comparisons to Pocket2Mol. I think overall the results are mixed and I wouldn't say that the proposed method is clearly always better than a corresponding autoregressive model (like Pocket2Mol). That said, I think the additional results do provide a better picture and are valuable and improve the paper. Hence, I raised my score. Everything considered, I feel still borderline about this paper, though, due to limited technical novelty and mixed or only marginally improved results compared to the relevant baselines. I do not have any further questions.

---

### Official Review · Reviewer_GWjC · 2022-10-25

**Confidence:** 4
**Correctness:** 4
**Technical Novelty And Significance:** 3
**Empirical Novelty And Significance:** 3
**Recommendation:** 6

**Clarity, Quality, Novelty And Reproducibility:**

**Clarity.** The paper is written in a clear way. The figures are clean and well support the model description and nicely illustrate the experimental findings. The training details, including pseudocode, are provided.

**Quality.** The model is described in full detail, and the experimental results are supported by many figures showing different aspects of the model. There are four recent models compared against TargetDiff, and multiple metrics are used. The main claims of the paper are proven in the appendix.

**Novelty.** Diffusion models are a new class of generative models, and this is one of the first examples of using them for target-based molecule generation. The concept of affinity predictor trained in an unsupervised fashion also seems novel.

**Reproducibility.** The code is not available, but the authors say they will publish the code when the paper is accepted. Based on the description, the reimplementation would be probably possible but tedious. Pseudocode is provided for the training and sampling procedure.

**Strength And Weaknesses:**

Strengths:
- Non-autoregressive sampling helps to better fill the available space in the binding pocket.
- The related work contains a set of very recent publications demonstrating the advancements in the fields of target-based molecule generation and diffusion models.
- SE(3)-equivariant networks ensure the strong performance independent of the protein orientation.
- Both continuous and discrete diffusion models are used to model coordinates and atom types, respectively.
- The main claims of the paper are formally proven.
- The authors discover a correlation between binding affinity and the entropy of the predicted atom representation when atom coordinates are frozen. Thus, this entropy can be treated as an affinity predictor trained in an unsupervised fashion.

Weaknesses:
- Currently, the code is not available, but it should be open-sourced upon publication.
- In Tables 3 and 4, confidence intervals could be added to make these results more convincing.

Questions:
- How do you combine v entropy and vina scores in Section 4.3?
- Coordinates and atom types are modeled independently in Equation 2. Do you think this could have a negative impact on the generated structures?

Minor points:
- Typos: “more structural data become available and unlock new opportunities”, “protien”->”protein”
- The full name “Center of Mass (CoM)” should be introduced at its first occurrence.
- Figure 7 reference is missing.


**Summary Of The Paper:**

In this paper, TargetDiff, a new 3D diffusion model, is introduced. The model generates molecules in a non-autoregressive fashion. The generated molecules are conditioned on the binding pocket, and the generation is equivariant to rotations and translations thanks to the equivariant GNNs used. Additionally, an unsupervised entropy-based method for ranking molecules is proposed. The model is evaluated against multiple recent generative models: liGAN, AR, GraphBP, and Pocket2Mol. The results indicate the strong performance of the proposed method.

**Summary Of The Review:**

Based on the above comments, I am leaning towards the acceptance of this paper.

---

> ### Author Response · Authors · 2022-11-14
> **Response to Reviewer GWjC**
>
> We thank the reviewer for the feedback and suggestions. Please see below for our responses to the comments.
>
> **Q1: Currently, the code is not available, but it should be open-sourced upon publication.**
>
> A1: Yes, we are committed to open sourcing the data as well as train/inference code upon publication.
>
> **Q2: In Tables 3 and 4, confidence intervals could be added to make these results more convincing.**
>
> A2: This is a great suggestion! We will add the confidence interval during the revision period instead due to the limited time in the rebuttal period.
>
> **Q3: How do you combine v entropy and vina scores in Section 4.3?**
>
> A3: We standardize the v entropy and vina scores separately (mean = 0, std = 1) to convert them in to the same scale (as $Z$-score), and then directly sum them up as a combined indicator.
>
> **Q4: Coordinates and atom types are modeled independently in Equation 2. Do you think this could have a negative impact on the generated structures?**
>
> A4: As illustrated in the paragraph under Equation 2, we choose to decompose the joint distribution as the product of coordinates and atom types for the a more efficient sampling during the **forward** phase. In the **generation** process, the dependencies between them are well modeled as Equation 5 shows, where the neural network takes inputs both noisy atom coordinates and atom types and predicts the denoised ones.

---

> > ### Comment · Reviewer_GWjC · 2022-11-23
> > **Thank you for your response**
> >
> > Thank you for your response. This answers my questions.

---

### Decision · Program_Chairs · 2023-01-20

**Decision:**

Accept: poster

**Justification For Why Not Higher Score:**

The novelty is still limited. The claim about NAT is not solid, according to its experimental results.

**Justification For Why Not Lower Score:**

The novel technical contribution is adequate to justify its appearance in ICLR.

**Metareview: Summary, Strengths And Weaknesses:**

The paper propose a diffusion model for molecule generation based on the target 3D structure. The model generates molecules in a non-autoregressive way. The generated molecules are conditioned on the target binding pocket, and the generation is equivariant to rotations and translations thanks to the equivariant GNNs used. Additionally, an unsupervised entropy-based method for ranking molecules is proposed. Experiments show promising performance against multiple recent generative models: liGAN, AR, GraphBP, and Pocket2Mol.

Strength of the paper:
1. The diffusion method based on target pocket 3D structure is a novel contribution.

 Weakness of the paper:
1. The benefits of non-autoregressive generation for molecules are not well supported by experimental results. Pocket2Mol is better. The motivation of NAT and claims in introduction should be adjusted.
2. The comparison with Hoogeboom et al should be elaborated.

Overall, after discussion, the reviewers agree that the paper has adequate interesting points to be included in the conference.


**Note From Pc:**

if the above contains the word "oral" or "spotlight" please see: "oral" presentation means -> notable-top-5% and "spotlight" means -> notable-top-25%. As stated in our emails, we are disassociating presentation type from AC recommendations

**Summary Of Ac-Reviewer Meeting:**

In the meeting, reviewers agree that the novel contribution (target-based 3D structure for diffusion) would be interesting to ICLR community. The claim about NAT should be adjusted.